# Instance-optimality in differential privacy via approximate inverse sensitivity mechanisms

**Hilal Asi**
Stanford University
asi@stanford.edu

**John C. Duchi**
Stanford University
jduchi@stanford.edu

## Abstract

We study and provide instance-optimal algorithms in differential privacy by extending and approximating the inverse sensitivity mechanism. We provide two approximation frameworks, one which only requires knowledge of local sensitivities, and a gradient-based approximation for optimization problems, which are efficiently computable for a broad class of functions. We complement our analysis with instance-specific lower bounds for vector-valued functions, which demonstrate that our mechanisms are (nearly) instance-optimal under certain assumptions and that minimax lower bounds may not provide an accurate estimate of the hardness of a problem in general: our algorithms can significantly outperform minimax bounds for well behaved instances. Finally, we use our approximation framework to develop private mechanisms for unbounded-range mean estimation, principal component analysis, and linear regression. For PCA, our mechanisms give an efficient (pure) differentially private algorithm with near-optimal rates.

## 1 Introduction

We study the estimation of a function (statistic) of interest under differential privacy, where strong privacy protections usually decrease utility relative to non-private data analysis. In an effort to improve the utility of private algorithms, it is of utmost importance to design mechanisms that adapt to the hardness of the underlying data. Such mechanisms are of growing prevalence in the privacy literature, with prominent examples including the smooth sensitivity [25] and propose-test-release [12] frameworks.

To further investigate adaptivity to underlying instance, Asi and Duchi [4] recently study instance-optimal mechanisms—which, in a sense, achieve optimal utility for every possible data instance—in differentially private release of 1-dimensional quantities, moving beyond the more standard (worst case) minimax optimality. Inspired by classical statistical theory, Asi and Duchi develop local-minimax optimality and optimality against unbiased mechanisms, both of which aim to capture the hardness of the underlying data. By developing instance-specific lower bounds, they show that classical frameworks such as smooth sensitivity and propose-test-release may not be instance-optimal in general. To overcome this challenge, they investigate what they term the *inverse sensitivity mechanism*, showing it is instance-optimal for a wide range of functions.

Yet instance-optimality in private statistical estimation remains widely unexplored. First, the implementation of the inverse sensitivity mechanism requires a calculation of a particular sample distance (see Section 1.1.1), which may be intractable. Moreover, the current instance-optimality guarantees are not sharp for vector-valued functions. This is in part because the paper [4] tailors its instance-optimality notions for 1-dimensional functions by leveraging Stein's "hardest one-dimensional alternative" approach to lower bounds [31, 9], which gives tight lower bounds for 1-dimensional functions but fails to yield correct bounds in higher dimensions.

To address these challenges, in this work we develop extensions and approximations to the inverse sensitivity mechanism with efficient implementations for a broad class of functions, which allows us to (for example) develop efficient algorithms for private PCA with near-optimal sample complexity. We also establish complementary instance-optimality results for vector-valued functions by proposing two approaches for instance-specific lower bounds: (i) a local minimax approach that measures the risk of an instance through the loss that an algorithm must incur on instances in a small neighborhood around it, and (ii) lower bounds against families of appropriately unbiased mechanisms, which includes many standard mechanisms. These instance-specific bounds suggest the limitations of more prevalent minimax (worst-case) bounds in privacy [19, 11]: they do not always give the correct limits on the performance of algorithms, and algorithms exist that achieve lower error on many instances.

## 1.1 Preliminaries

Given a function $f : \mathcal{X}^n \to \mathcal{T}$ and instance $\boldsymbol{x} \in \mathcal{X}^n$, we wish to design differentially private mechanisms that accurately estimates the value $f(\boldsymbol{x})$. We usually take $\mathcal{X}, \mathcal{T} \subset \mathbb{R}^d$ for a dimension $d$.

We begin by recalling the standard definition of differential privacy [16, 15]. We say that two instances $\boldsymbol{x}, \boldsymbol{x}' \in \mathcal{X}^n$ are *neighboring* if they differ in at most one example, that is, $d_{\text{ham}}(x, x') \leq 1$.

**Definition 1.1.** *A randomized algorithm* $M : \mathcal{X}^n \to \mathcal{T}$ *is* $(\varepsilon, \delta)$*-differentially private if for all neighboring datasets* $\boldsymbol{x}, \boldsymbol{x}' \in \mathcal{X}^n$ *and all measurable* $S \subseteq \mathcal{T}$,

$$\mathbb{P}\left(M(\boldsymbol{x}) \in S\right) \leq e^\varepsilon \mathbb{P}\left(M(\boldsymbol{x}') \in S\right) + \delta.$$

*If* $\delta = 0$, *then* $M$ *is* $\varepsilon$*-differentially private.*

Given a loss function $L : \mathcal{T} \times \mathcal{T} \to \mathbb{R}_+$, we quantify the utility of a mechanism $M$ on instance $\boldsymbol{x}$ through its expected loss $\mathbb{E}[L(M(\boldsymbol{x}), f(\boldsymbol{x}))]$. A mechanism is instance-optimal if it achieves the best utility for every instance. We formalize this through instance-specific lower bounds in Section 3.

For a function $f : \mathcal{X}^n \to \mathbb{R}$, the standard method to preserve privacy is the Laplace mechanism [16]. Defining the global sensitivity of $f$ to be $\text{GS}_f := \sup_{\boldsymbol{x}, \boldsymbol{x}' : d_{\text{ham}}(\boldsymbol{x}, \boldsymbol{x}') \leq 1} |f(\boldsymbol{x}) - f(\boldsymbol{x}')|$, it adds Laplace noise, $M_{\text{Lap}}(\boldsymbol{x}) := f(\boldsymbol{x}) + \frac{\text{GS}_f}{\varepsilon} \text{Lap}(1)$. This can be conservative, therefore Nissim et al. [25] consider the local sensitivity at instance $\boldsymbol{x}$ at hand $\text{LS}_f(\boldsymbol{x}) := \sup_{\boldsymbol{x}' : d_{\text{ham}}(\boldsymbol{x}, \boldsymbol{x}') \leq 1} |f(\boldsymbol{x}) - f(\boldsymbol{x}')|$. Directly using the local sensitivities may compromise privacy, hence the smooth sensitivity framework adds noise that is proportional to a smooth upper bound $S^\beta(\boldsymbol{x})$ on the local sensitivity, that is, $M_{\text{sm}}(\boldsymbol{x}) := f(\boldsymbol{x}) + \frac{2S^\beta(\boldsymbol{x})}{\varepsilon} Z$, where $Z$ is sampled from an admissible noise distribution and $S^\beta(\boldsymbol{x})$ is the smooth sensitivity satisfying $\text{LS}(\boldsymbol{x}) \leq S^\beta(\boldsymbol{x})$ and $S^\beta(\boldsymbol{x}) \leq e^\beta S^\beta(\boldsymbol{x}')$ for neighboring instances $\boldsymbol{x}, \boldsymbol{x}'$, and $\beta$ is chosen appropriately to guarantee the desired privacy level.

### 1.1.1 The inverse sensitivity mechanism

Our work builds on the inverse sensitivity mechanism [4], which we review. Key to the mechanism is the path-length (inverse sensitivity), which, for a target $t$, measures how many users we must change in $\boldsymbol{x}$ to reach $\boldsymbol{x}'$ with a target value $t$:

$$\text{len}_f(\boldsymbol{x}; t) := \inf_{\boldsymbol{x}'} \left\{ d_{\text{ham}}(\boldsymbol{x}, \boldsymbol{x}') \mid f(\boldsymbol{x}') = t \right\}. \tag{1}$$

The basic inverse sensitivity mechanism then instantiates the exponential mechanism [24] with the path-length function (1), yielding the density

$$\pi_{M_{\text{inv}}(\boldsymbol{x})}(t) = \frac{e^{-\text{len}_f(\boldsymbol{x}; t)\varepsilon/2}}{\int_\mathcal{T} e^{-\text{len}_f(\boldsymbol{x}; s)\varepsilon/2} ds}. \tag{M.1}$$

A smoother variant of mechanism (M.1) is sometimes necessary to achieve instance-optimality, where one instead uses

$$\text{len}_f^\rho(\boldsymbol{x}; t) = \inf_{s \in \mathcal{T} : \|s - t\| \leq \rho} \text{len}_f(\boldsymbol{x}; s),$$

with a smoothing parameter $\rho > 0$ [4]. Different variations of these mechanisms are instance-optimal for a range of real-valued functions. Yet while examples exist, it is often unclear how to compute the length (1).

Instance-specific bounds depend on the *modulus of continuity*, which (focusing in this work on a vector space $\mathcal{T}$ with norm $\|\cdot\|_p$) measures the sensitivity of a function when changing $k$ users:

$$\omega_f^p(\boldsymbol{x}; k) = \sup_{\boldsymbol{x}' \in \mathcal{X}^n} \left\{ \|f(\boldsymbol{x}) - f(\boldsymbol{x}')\|_p : d_{\mathrm{ham}}(\boldsymbol{x}, \boldsymbol{x}') \leq k \right\}. \tag{2}$$

Instance-specific lower bounds show that the risk we expect for an $\varepsilon$-differentially private algorithm on instance $\boldsymbol{x}$ is in general roughly $\omega_f(\boldsymbol{x}; 1/\varepsilon)$ for 1-dimensional functions (with $p = 1$) and loss $L(s, t) = |s - t|$ [4]. Unfortunately, this is not tight for $d$-dimensional functions.

**Notation** We denote samples using bold symbols $\boldsymbol{x} \in \mathcal{X}^n$ and individual examples using non-bold symbol $x \in \mathcal{X}$. We let $d_{\mathrm{ham}}(\boldsymbol{x}, \boldsymbol{x}')$ denote the Hamming distance of instance $\boldsymbol{x}, \boldsymbol{x}' \in \mathcal{X}^n$. The local sensitivity of $f : \mathcal{X}^n \to \mathcal{T}$ at instance $\boldsymbol{x}$ is $\mathrm{LS}_f^p(\boldsymbol{x}) = \omega_f^p(\boldsymbol{x}; 1)$, and the global sensitivity of $f$ is $\mathrm{GS}_f^p = \sup_{\boldsymbol{x} \in \mathcal{X}^n} \omega_f^p(\boldsymbol{x}; 1)$. To facilitate notation, we sometimes remove the superscript $p$ if $p = 2$. We let $\mathrm{diam}_p(\mathcal{T}) = \sup_{s,t \in \mathcal{T}} \|s - t\|_p$, and $\mathbb{B}_p^{d-1} = \{x \in \mathbb{R}^d : \|x\|_p \leq 1\}$ denote the $\ell_p$-ball.

## 1.2 Contributions

**Approximate inverse sensitivity mechanisms** We develop two approximation methods for the inverse sensitivity mechanism: (i) using local sensitivities in Section 2.1 and (ii) a gradient-based method for minimization problems in Section 2.2. These methods have efficient implementations for a wide range of problems and can outperform smooth sensitivity mechanisms for pure differential privacy. In contrast to Cauchy and Student's T distributions used in such instantiations [7]—which have infinite first and third moment respectively—our mechanisms add noise with bounded $p$'th moments for all finite $p$, resulting in improved high-probability bounds for utility analysis which is especially important for high dimensional functions as our examples demonstrate.

**Instance-optimality and lower bounds** We propose two notions of instance optimality for vector-valued functions and prove tight lower bounds for both notions in Section 3. Similarly to the 1-dimensional setting, our results give a characterization of the risk through the modulus of continuity. Combined with our instance-specific upper bounds, these bounds establish that approximate and exact inverse mechanisms are (nearly) instance-optimal for vector-valued functions under some assumptions.

**Applications** We study three problems that illustrate the methodological possibilities of the inverse sensitivity framework and its approximations in Section 4: mean estimation, PCA and linear regression. The utility improvements in these examples demonstrate the advantages of our mechanisms over standard frameworks and the importance of these notions of instance-optimality. Here we highlight the PCA example where smooth sensitivity algorithms require sample complexity (for dimension $d$ and ignoring other parameters) $O(d^{3/2})$ [18], whereas our mechanisms require $O(d)$ samples, which is the optimal dependence on the dimension $d$ according to PCA lower bounds [10, 21].

## 1.3 Related work

The most widely used frameworks for instance-dependent noise are smooth sensitivity [25] and propose-test-release [12]. The former adds noise that scaling with a smooth upper bound on the local sensitivity, and the latter adds noise scaling with a prespecified upper bound on the local sensitivity—whose validity the algorithm tests—in a neighborhood of the instance. Applications are numerous: Smith and Thakurta [30] develop an algorithm based on propose-test-release for high-dimensional regression problems, and Bun and Steinke [7] design noise distributions for smooth sensitivity and use them to estimate the mean of distributions with unbounded range. Other applications include principal component analysis [18], outlier analysis [26], and graph data [22, 32]. The inverse sensitivity framework is a distinct approach to instance-dependent noise that Asi and Duchi [4] investigate ([20, 29, 8] propose variants of the mechanism). Their results suggest that this framework, in contrast to smooth sensitivity and propose-test-release, is instance-optimal for a range of functions, and can have quadratically better sample complexity than smooth sensitivity mechanisms.

## 2 Approximate inverse sensitivity mechanisms

Having described the difficulty of sampling from the inverse sensitivity mechanism in general, in this section we develop two approximation frameworks that are applicable for a broader range of functions while maintaining some of the instance-optimality guarantees of the exact mechanism. First, in Section 2.1 we describe a method that uses the local sensitivities to approximate the path-length, and in Section 2.2 we describe an approximation for the specific setting of empirical risk minimization.

### 2.1 Approximation using local sensitivities

The mechanisms we develop in this section first construct an approximation $\overline{\mathsf{len}}_f(\boldsymbol{x}; t)$ for the path-length, then apply the exponential mechanism for a base measure $\mu$ on $\mathcal{T}$ with this approximation

$$\pi_{M_{\mathrm{appr}}(\boldsymbol{x})}(t) = \frac{e^{-\overline{\mathsf{len}}_f(\boldsymbol{x};t)\varepsilon/2}}{\int_{\mathcal{T}} e^{-\overline{\mathsf{len}}_f(\boldsymbol{x};s)\varepsilon/2} d\mu(s)}. \tag{M.2}$$

Our main tool for calculating $\overline{\mathsf{len}}_f(\boldsymbol{x}; t)$ are the local sensitivities of instance $\boldsymbol{x}$ at distance $\ell$

$$\mathrm{LS}_\ell^p(\boldsymbol{x}) = \sup_{\boldsymbol{x}': d_{\mathrm{ham}}(\boldsymbol{x}, \boldsymbol{x}') = \ell} \mathrm{LS}^p(\boldsymbol{x}').$$

This definition implies that changing $k$ users can vary the function value by at most $\sum_{\ell=1}^k \mathrm{LS}_\ell^p(\boldsymbol{x})$. As a consequence, we have the lower bound $\mathsf{len}_f(\boldsymbol{x}; t) \geq \min\{k : \sum_{i=1}^k \mathrm{LS}_i^p(\boldsymbol{x}) \geq \|t - f(\boldsymbol{x})\|_p\}$. Unfortunately, directly using this lower bound may result in mechanisms that are not private. The following theorem shows how to construct suitable approximations that preserve privacy.

**Theorem 1.** *Let $f : \mathcal{X}^n \to \mathcal{T}$ and $R_\ell : \mathcal{X}^n \to \mathbb{R}$ satisfy $\mathrm{LS}^p(\boldsymbol{x}) \leq R_1(\boldsymbol{x})$ and $R_\ell(\boldsymbol{x}) \leq R_{\ell+1}(\boldsymbol{x}')$ for any neighboring instances $\boldsymbol{x}, \boldsymbol{x}' \in \mathcal{X}^n$. Then, using the approximation*

$$\overline{\mathsf{len}}_f(\boldsymbol{x}; t) = \min \left\{ k : \sum_{i=1}^k R_i(\boldsymbol{x}) \geq \|t - f(\boldsymbol{x})\|_p \right\}, \tag{3}$$

*mechanism (M.2) is $\varepsilon$-differentially private.*

Algorithm 1 efficiently samples from the approximate inverse sensitivity mechanism for reasonable choices of $p$: the main bottleneck is step 3 but efficient algorithms exist for $p \in \{1, 2\}$ using truncated Gamma distributions [23].

---

**Algorithm 1:** Sampling from approximate inverse sensitivity

---

**Input:** $\boldsymbol{x} \in \mathcal{X}^n$, $p$, $\{R_i(\cdot)\}_{i=1}^n$

1 Denote $S_k = \{t : \sum_{\ell=1}^{k-1} R_\ell(\boldsymbol{x}) \leq \|t\|_p \leq \sum_{\ell=1}^k R_\ell(\boldsymbol{x})\}$;

2 Sample $k \sim K$ from $\mathbb{P}(K = k) \propto \mathrm{Vol}(S_k) e^{-k\varepsilon/2}$ for $1 \leq k \leq n$;

3 Sample $z \sim \mathsf{Uni}(S_k)$;

4 **return** $f(\boldsymbol{x}) + z$

---

Before proceeding to our utility analysis, we show that given the local sensitivities, we can always find an appropriate choice of $R_i$ without calculating the smooth sensitivities.

**Proposition 2.1.** *Let $f : \mathcal{X}^n \to \mathbb{R}^d$ and assume $\overline{\mathrm{LS}}(\boldsymbol{x})$ is such that $\mathrm{LS}(\boldsymbol{x}) \leq \overline{\mathrm{LS}}(\boldsymbol{x})$ for every $\boldsymbol{x}$. Then mechanism (M.2) using the approximation (3) with $R_\ell(\boldsymbol{x}) = \sup_{\boldsymbol{x}': d_{\mathrm{ham}}(\boldsymbol{x}, \boldsymbol{x}') \leq \ell} \overline{\mathrm{LS}}(\boldsymbol{x})$ is $\varepsilon$-differentially private.*

#### 2.1.1 Utility guarantees for vector-valued functions

In this section, we provide utility guarantees for the exact and approximate inverse sensitivity mechanisms for vector-valued functions. Combined with our lower bounds of Section 3, this establishes (near) instance optimality of these methods. Our guarantees hold with high probability, in contrast to those of the smooth sensitivity framework which uses distributions with heavy tails. We also show that our approximations can outperform smooth Laplace for real-valued functions.

We begin by analyzing the utility of the exact and approximate inverse sensitivity mechanisms.

**Theorem 2.** *Let $f : \mathcal{X}^n \to \mathbb{R}^d$, $\mathrm{diam}_2(f(\mathcal{X}^n)) \leq D$, $r > 0$ and $1 \leq K \leq n$. Then the (smooth) inverse sensitivity mechanism* (M.1) *with $\rho = 1/n^r$ has*

$$\mathbb{P}\left(\|M_{\mathrm{inv}}(\boldsymbol{x}) - f(\boldsymbol{x})\|_2 \geq \omega_f(\boldsymbol{x}; K) + 1/n^r\right) \leq e^{-K\varepsilon/2}(n^r D)^d.$$

*Moreover, if $R_i(\boldsymbol{x}) \leq D$, the approximate mechanism* (M.2) *using* (3) *with $p = 2$ has*

$$\mathbb{P}\left(\|M_{\mathrm{appr}}(\boldsymbol{x}) - f(\boldsymbol{x})\|_2 \geq \sum_{i=1}^{K} R_i(\boldsymbol{x})\right) \leq e^{-K\varepsilon/2+1}\left(nD\bigg/\sum_{i=1}^{K} R_i(\boldsymbol{x})\right)^d.$$

We remark that using the smooth sensitivity framework to preserve pure differential privacy does not usually result in such high probability bounds due to using noise distributions with heavy tails such as Cauchy distribution [25]. Moreover, Theorem 2 implies that using $k \approx \frac{Cd\log n}{\varepsilon}$ for large constant $C$, with high probability the inverse sensitivity mechanism roughly has

$$\|M_{\mathrm{inv}}(\boldsymbol{x}) - f(\boldsymbol{x})\|_2 \leq O(\omega_f(\boldsymbol{x}; Cd\log n/\varepsilon)).$$

The approximate mechanism has similar loss whenever our approximate $R_i$ are accurate such that $\sum_{i=1}^{K} R_i(\boldsymbol{x}) = O(\omega_f(\boldsymbol{x}; K))$. The lower bounds in Section 3 show this is (near) instance optimal.

We conclude this section with another choice of $R_i$ that uses the smooth sensitivities instead. This guarantees that the approximate mechanism always outperforms the smooth Laplace mechanism [25].

**Proposition 2.2.** *Let $f : \mathcal{X}^n \to \mathbb{R}$, $\varepsilon = O(1)$ and $R_\ell(\boldsymbol{x}) = \sup_{\boldsymbol{x}':d_{\mathrm{ham}}(\boldsymbol{x},\boldsymbol{x}')\leq\ell} \mathrm{S}^\beta(\boldsymbol{x}')$. Then mechanism* (M.2) *using* (3) *is $\varepsilon$-differentially private. If $p = 1$ and $\beta = \frac{\varepsilon}{8}$ then $\mathbb{E}\left[|M_{\mathrm{appr}}(\boldsymbol{x}) - f(\boldsymbol{x})|\right] \leq O(\frac{\mathrm{S}^\beta(\boldsymbol{x})}{\varepsilon})$.*

The smooth Laplace mechanism—which guarantees only approximate $(\varepsilon, \delta > 0)$-DP—has loss $O(\frac{\mathrm{S}^\beta(\boldsymbol{x})}{\varepsilon})$ with a much smaller $\beta = \frac{\varepsilon}{2\log 2/\delta}$, which can be $\log 1/\delta$ worse in some settings.

## 2.2 Gradient-based approximations for empirical risk minimization

In this section, we describe our second approximation which applies to empirical risk minimization problems. Given data points $(x_i, y_i) \in \mathbb{R}^d \times \mathbb{R}$ and $L$-Lipschitz loss function $\ell(\theta; x_i)$ for $\theta \in \Theta$, we wish to solve the following minimization problem

$$\hat{\theta}_n = \operatorname*{argmin}_{\theta \in \Theta} L_n(\theta; \boldsymbol{x}, \boldsymbol{y}) \coloneqq \frac{1}{n} \sum_{i=1}^{n} \ell(\theta; x_i, y_i).$$

It is possible to calculate the path-length using gradients for robust regression [4]. Here, we use similar techniques to approximate the inverse sensitivity mechanism in general settings. As $\ell$ is $L$-Lipschitz, we need to change $\mathsf{len}(\boldsymbol{x}, \boldsymbol{y}; \theta) \geq \frac{n}{L}\|\nabla L_n(\theta; \boldsymbol{x}, \boldsymbol{y})\|_2$ users to make $\theta$ a minimizer with $\nabla L_n(\theta; \boldsymbol{x}', \boldsymbol{y}') = 0$. The *gradient mechanism* uses this approximation of $\mathsf{len}$, resulting in the density

$$\pi_{\mathsf{Grad}}(\theta \mid \boldsymbol{x}, \boldsymbol{y}) \propto e^{-\frac{n\varepsilon}{2L}\|\nabla L_n(\theta; \boldsymbol{x}, \boldsymbol{y})\|_2}. \tag{4}$$

Sampling from this distribution can be hard in general, but we show an efficient implementation for linear regression in Section 4.3. For general twice differentiable functions, we propose an efficient heuristic of the gradient mechanism based on Taylor's expansion which gives $\nabla L_n(\theta; \boldsymbol{x}, \boldsymbol{y}) \approx \nabla^2 L_n(\hat{\theta}_n; \boldsymbol{x}, \boldsymbol{y})(\theta - \hat{\theta}_n)$. Letting $\mathsf{GS}_{\mathsf{Hess}}$ denote the global sensitivity of $\|\nabla^2 L_n(\hat{\theta}_n; \boldsymbol{x}, \boldsymbol{y})(\theta - \hat{\theta}_n)\|_2$, we define the following *Hessian-based mechanism* for $\theta \in \Theta$

$$\pi_{\mathsf{Hess}}(\theta \mid \boldsymbol{x}, \boldsymbol{y}) \propto e^{-\frac{\varepsilon}{2\mathsf{GS}_{\mathsf{Hess}}}\|\nabla^2 L_n(\hat{\theta}_n; \boldsymbol{x}, \boldsymbol{y})(\theta - \hat{\theta}_n)\|_2}. \tag{5}$$

The main advantage of the Hessian mechanism (5) is that now we can design efficient and simple sampling procedures (see Section 4.3). It also provides an accurate approximation of the gradient mechanism with good utility whenever $\nabla^2 L_n$ is $H$-Lipschitz with small $H$.

The privacy of these mechanisms follow immediately from the privacy of the exponential mechanism. For utility, we start with the following lemma which upper bound $\mathsf{GS}_{\mathsf{Hess}}$.

**Lemma 2.1.** *Assume $\ell(\cdot; x_i, y_i)$ is $L$-Lipschitz, $\nabla^2 L_n(\hat{\theta}_n; \boldsymbol{x}, \boldsymbol{y}) \succeq \lambda I$, $\hat{\theta}_n \in \text{int}\,\Theta$, and $\nabla^2 L_n(\cdot; \boldsymbol{x}, \boldsymbol{y})$ is $H$-Lipschitz. If $H \le \lambda = O(1)$ and $n \ge 4L \operatorname{diam}_2(\Theta) + 1$ then $\mathsf{GS}_{\mathsf{Hess}} \le O\left(\frac{L}{n}\right)$.*

We are now ready to analyze the utility of the Hessian-based mechanism.

**Proposition 2.3.** *Let the set of instances $(x_i, y_i)_{i=1}^n$ satisfy the assumptions of Lemma 2.1. Then the Hessian mechanism (5) is $\varepsilon$-DP. If $\inf_{\theta \in \text{bd}\,\Theta} \|\theta - \hat{\theta}_n\|_2 \ge \Omega(\frac{dL^2}{n^2 \varepsilon^2} \operatorname{tr}(\nabla^2 L(\hat{\theta}_n; \boldsymbol{x}, \boldsymbol{y})^{-2}))$, then*

$$\mathbb{E}_{\theta \sim \pi_{\mathsf{Hess}}(\cdot|\boldsymbol{x},\boldsymbol{y})} \left[ \|\theta - \hat{\theta}_n\|_2^2 \right] \le O\left( \frac{dL^2 \operatorname{tr}(\nabla^2 L(\hat{\theta}_n; \boldsymbol{x}, \boldsymbol{y})^{-2})}{n^2 \varepsilon^2} \right).$$

When $H = 0$, the gradient (4) and Hessian mechanisms (5) are identical and the gradient mechanism has the same utility. In Section 4.3, we use these mechanisms for solving regression problems and show the significant advantages of instance-specific bounds over standard minimax bounds.

Finally, we remark that our gradient-based approximations of the inverse sensitivity mechanism are closely related to the K-norm mechanism [27] where the authors use gradient norms as a score function for the exponential mechanism. However, their work only provides asymptotic utility analyses without finite-sample guarantees, and they propose an approximate implementation of their mechanisms using an MCMC procedure without providing privacy guarantees for the implementation.

# 3 Instance-specific lower bounds for vector-valued functions

Given a function $f : \mathcal{X}^n \to \mathbb{R}^d$, in this section we prove instance-specific lower bounds on the loss that any private mechanism must incur. Unfortunately, the instance-specific notions in [4] were tailored for 1-dimensional functions, hence do not result in satisfactory lower bounds in our setting. To this end, we propose extensions that result in tight bounds. The first notion gives lower bounds by restricting to families of appropriately unbiased mechanisms. The second is a local-minimax approach that measures the performance in a small neighborhood around a given instance.

We begin with our optimality notion for unbiased mechanisms, which we define now.

**Definition 3.1.** *We say that a randomized algorithm $M$ is $\|\cdot\|$-unbiased if for any $\boldsymbol{x}, \boldsymbol{x}' \in \mathcal{X}^n$ and $\rho > 0$,*

$$\mathbb{P}(\|M(\boldsymbol{x}) - f(\boldsymbol{x})\| \le \rho) \ge \mathbb{P}(\|M(\boldsymbol{x}) - f(\boldsymbol{x}')\| \le \rho).$$

Definition 3.1 says that when applying an unbiased mechanism $M$ on instance $\boldsymbol{x}$, the output is more likely to be in a ball around the correct value $f(\boldsymbol{x})$ rather than $f(\boldsymbol{x}')$ for some other instance $\boldsymbol{x}'$. Anderson's theorem [2] implies that the Laplace mechanism, Gaussian mechanism, their smooth sensitivity instantiations, the approximate inverse sensitivity mechanism, and any instantiation of the exponential mechanism with a concave score function are $\|\cdot\|$-unbiased.

Our lower bounds require a growth condition on the set of values at distance at most $k$, $W_f(\boldsymbol{x}; k) = \{f(\boldsymbol{x}') : d_{\text{ham}}(\boldsymbol{x}, \boldsymbol{x}') \le k\}$. We have the following instance-specific lower bound for unbiased mechanisms.

**Theorem 3.** *Let $f : \mathcal{X}^n \to \mathbb{R}^d$ and assume $W_f(\boldsymbol{x}; k) \supseteq c \cdot \omega_f(\boldsymbol{x}; k) \cdot \mathbb{B}_2^{d-1}$ for $c > 0$. If $M$ is $\varepsilon$-DP,*

$$\sup_{\boldsymbol{x} \in \mathcal{X}^n} \mathbb{E}\left[\|M(\boldsymbol{x}) - f(\boldsymbol{x})\|_2\right] \ge \frac{c}{8} \sup_{\boldsymbol{x} \in \mathcal{X}^n} \max_{1 \le k \le n} e^{-k\varepsilon/d} \omega_f(\boldsymbol{x}; k).$$

*Moreover, if $M$ is $\|\cdot\|_2$-unbiased, then for any $\boldsymbol{x} \in \mathcal{X}^n$,*

$$\mathbb{E}\left[\|M(\boldsymbol{x}) - f(\boldsymbol{x})\|_2\right] \ge \frac{c}{8} \max_{1 \le k \le n} e^{-2k\varepsilon/d} \omega_f(\boldsymbol{x}; k).$$

Theorem 3 suggests that worst-case lower bounds may be too pessimistic: while the minimax risk is roughly $\sup_{\boldsymbol{x} \in \mathcal{X}^n} \omega_f(\boldsymbol{x}; d/\varepsilon)$, we may hope to achieve a better risk for instance $\boldsymbol{x}$, that is, $\omega_f(\boldsymbol{x}; d/\varepsilon)$.

Now we define the local-minimax risk for an instance $\boldsymbol{x}$ following similar ideas in statistical theory [cf. 33, Ch. 8]. Let $\mathcal{M}_\varepsilon$ be the family of $\varepsilon$-differentially private mechanisms. For a radius $r$, we define the local-minimax risk of $\boldsymbol{x}$ to be the worst-case risk in a small neighborhood around $\boldsymbol{x}$, that is,

$$\mathcal{R}(\boldsymbol{x}; r) := \inf_{M \in \mathcal{M}_\varepsilon} \sup_{\boldsymbol{x}' : d_{\text{ham}}(\boldsymbol{x}, \boldsymbol{x}') \le r} E\left[\|M(\boldsymbol{x}') - f(\boldsymbol{x}')\|_2\right]. \tag{6}$$

The choice of $r$ in this definition is important to exclude trivial mechanisms such as $M(\boldsymbol{x}') = f(\boldsymbol{x})$; briefly, we choose the smallest radius that excludes such mechanisms, which is in this case $r = \Theta(d/\varepsilon)$ (see Appendix C.2 for more details about this definition).

We have the following lower bound for the local-minimax risk.

**Theorem 4.** *Let $f : \mathcal{X}^n \to \mathbb{R}^d$ and assume $W_f(\boldsymbol{x}; k) \supseteq c \cdot \omega_f(\boldsymbol{x}; k) \cdot \mathbb{B}_2^{d-1}$ for $c > 0$. Then for any $\boldsymbol{x} \in \mathcal{X}^n$, $\mathcal{R}(\boldsymbol{x}; d/\varepsilon) \geq \Omega\left(\omega_f(\boldsymbol{x}; d/\varepsilon)\right).$*

Similarly to our lower bounds for unbiased mechanisms, Theorem 4 shows that any mechanism must incur local-minimax risk roughly $\omega_f(\boldsymbol{x}; d/\varepsilon)$ for instance $\boldsymbol{x}$. The upper bounds of Theorem 2 show that the exact inverse mechanism achieves this loss for every instance up to logarithmic factors, as well as the approximate version if the approximations $R_i$ are accurate.

# 4 Applications

We investigate three examples that demonstrate different advantages and applications of the exact, approximate, and gradient inverse sensitivity mechanisms. Our examples include (i) mean estimation with unbounded range, (ii) principal component analysis and (iii) linear regression, and show that our techniques yield private algorithms with better noise distributions resulting in improved utility, which in some cases can significantly outperform existing minimax-optimal algorithms.

## 4.1 Unbounded-range mean estimation

Given $x_i \overset{\text{iid}}{\sim} P$ with unbounded range, our goal is to privately estimate the mean $\mu = \mathbb{E}_{x \sim P}[x]$. The difficulty here is that the empirical mean has infinite global and even local sensitivity, leading Bun and Steinke [7] to use the trimmed mean which calculates the mean after removing the smallest and largest $m$ samples. Letting $x_{(1)} \leq \cdots \leq x_{(n)}$ denote the order statistics, the trimmed mean is

$$\mathsf{trim}_m(\boldsymbol{x}) = \frac{x_{(m+1)} + x_{(m+2)} + \cdots + x_{(n-m)}}{n - 2m}. \tag{7}$$

This is useful as it leads to small local sensitivity under distributional assumptions. Bun and Steinke [7] use the smooth sensitivity to estimate the trimmed mean, resulting in strong utility but only with the weaker concentrated differential privacy [14, 6]. To preserve pure differential privacy, they use Student's T distribution which has infinite third moments and consequently heavy tails.

We use the exact inverse mechanism to estimate the mean with strong utility. This algorithm has finite $p$'th moment for any finite $p$, therefore yields tight confidence intervals. We assume $\mu \in [a, b]$ and let $[c]_{[a,b]}$ denote projection to $[a, b]$. The following lemma enables exact calculation of the path-length.

**Lemma 4.1.** *Let $f(\boldsymbol{x}) = [\mathsf{trim}_m(\boldsymbol{x})]_{[a,b]}$. Then, for any $t \in [a, b]$, if $t \geq \mathsf{trim}_m(\boldsymbol{x})$, we have $\mathsf{len}_f(\boldsymbol{x}; t) = \min\{k : k \leq m, t - f(\boldsymbol{x}) \leq \frac{1}{n-2m} \sum_{i=1}^{k} (x_{(n-m+i)} - x_{(m+i)})\} \cup \{m + 1\}$.*

The calculation for $t < \mathsf{trim}_m(\boldsymbol{x})$ is similar and we present it in Appendix D. Using Lemma 4.1, we can efficiently sample (Algorithm 4 in Appendix D.1 which runs in $O(n \log n)$ time) from the inverse sensitivity mechanism. To analyze the performance of this algorithm, we assume $P$ is $\sigma$-subgaussian while noting that these results can be extended to other settings in [7]. The following proposition upper bounds the error of our algorithm, which resembles the bounds that the algorithms of [7] achieve with the weaker concentrated differential privacy.

**Proposition 4.1.** *Let $a, b \in \mathbb{R}$ and $x_i \overset{\text{iid}}{\sim} P$ where $P$ is $\sigma$-subgaussian with mean $\mu \in [a, b]$. If $P$ is symmetric about its mean and $n \geq \frac{12 \log(n(b-a)/\sigma^2)}{\varepsilon}$, the inverse sensitivity mechanism (Algorithm 4) with $\rho = \frac{\sigma^2}{n^2}$ is $\varepsilon$-differentially private and has*

$$\mathbb{E}[(\hat{x} - \mu)^2] \leq \frac{\sigma^2}{n} + \frac{\sigma^2}{n^2} \cdot O\left(\frac{\log((b-a)/\sigma)}{\varepsilon} + \frac{\log n}{\varepsilon^2}\right).$$

## 4.2 Principal component analysis

In this section, we apply our approximations to calculate a rank $k$ approximation of a matrix. Given $x_1, \ldots, x_n \in \mathbb{B}_2^{d-1}$ with covariance $\Sigma(\boldsymbol{x}) = \frac{1}{n} \sum_{i=1}^n x_i x_i^T$, we wish to find $V \in \mathbb{R}^{d \times k}$ that solves

$$\hat{V}(\boldsymbol{x}) = \underset{V : V^T V = I_k}{\operatorname{argmax}} \; F(V) := \operatorname{tr}(V^T \Sigma(\boldsymbol{x}) V) \tag{8}$$

Gonen and Gilad-Bachrach [18] design $\varepsilon$-DP algorithms based on the smooth sensitivity framework with suboptimal error of roughly $\frac{d^{3/2}}{n\mathsf{GAP}(\boldsymbol{x})\varepsilon}$. We show that our algorithms achieve a near-optimal rate $\frac{d}{n\mathsf{GAP}(\boldsymbol{x})\varepsilon}$. Though there exist algorithms that achieve this rate using the exponential mechanism [10, 21], these algorithms require sampling from complex distributions and the only implementation with theoretical runtime analysis requires $O(d^6)$ time. In contrast, given the eigenvectors, our algorithm (Algorithm 2) returns a private version of the leading eigenvector in time $O(n + d)$ with high probability.

We only consider $k = 1$ as extensions to larger $k$ are straightforward using QR factorization [18]. Our algorithm builds on techniques from [18] and the approximate inverse mechanism. It requires a non-private PCA algorithm $\mathcal{A}_1$ that calculates the first eigenvector $\hat{v} \in \mathbb{R}^d$ (which maximizes (8)) and the gap between the two largest eigenvalues $\mathsf{GAP}(\boldsymbol{x}) := \lambda_1(\Sigma(\boldsymbol{x})) - \lambda_2(\Sigma(\boldsymbol{x}))$. Then it randomly flips the sign of $\hat{v}$ as $-\hat{v}$ is also a solution, and adds noise using the approximate inverse sensitivity. Algorithm 2 describes our private PCA procedure. Given the output of the non-private PCA algorithm, the main computational difficulty in Algorithm 2 is step 3 which requires sampling from the noise distribution of Algorithm 1. Using the rejection-sampling algorithms of Laud et al. [23] for sampling from truncated Gamma distributions (which has constant success probability in our setting), we can efficiently sample from Algorithm 1 in time $O(n + d)$ with high probability.

---

**Algorithm 2:** Private PCA using approximate inverse sensitivity

**Input:** $x$
1. Calculate $\hat{v} = \mathcal{A}_1(\boldsymbol{x}), \mathsf{GAP} = \mathsf{GAP}(\boldsymbol{x})$;
2. Set $\overline{v} = B\hat{v}$ for $B \sim \mathsf{Uni}\{-1, +1\}$;
3. Sample $z$ from (M.2) using Algorithm 1 with $p = 2$, $R_i = \min(C_{\mathsf{pca}}/(n\mathsf{GAP} - 2k), \sqrt{2})$;
4. **return** $v_{\mathsf{out}} = \frac{\overline{v} + z}{\|\overline{v} + z\|_2}$ ;

---

Following [18], we define the local sensitivity ($k = 1$) while taking into consideration the vector sign

$$\mathsf{LS}(\boldsymbol{x}) = \sup_{\boldsymbol{x}' : d_{\mathsf{ham}}(\boldsymbol{x}, \boldsymbol{x}') \leq 1} \min(\|\hat{V}(\boldsymbol{x}) - \hat{V}(\boldsymbol{x}')\|_2, \|\hat{V}(\boldsymbol{x}) + \hat{V}(\boldsymbol{x}')\|_2).$$

We build on the following key lemma that bounds the local sensitivity.

**Lemma 4.2** ([18], Theorem 5, Lemma 11). *If $\mathsf{GAP}(\boldsymbol{x}) > 0$ then there is a universal constant $C_{\mathsf{pca}} < \infty$ such that $\mathsf{LS}(\boldsymbol{x}) \leq \min(\frac{C_{\mathsf{pca}}}{n\mathsf{GAP}(\boldsymbol{x})}, \sqrt{2})$. Moreover, $|\mathsf{GAP}(\boldsymbol{x}) - \mathsf{GAP}(\boldsymbol{x}')| \leq 2d_{\mathsf{ham}}(\boldsymbol{x}, \boldsymbol{x}')/n$.*

Using this bound and the guarantees of our approximate mechanism, we get the following proposition.

**Proposition 4.2.** *Assume $n \geq 1/C_{\mathsf{pca}}$, $\beta > 0$ and $\Omega(\frac{d}{\mathsf{GAP}(\boldsymbol{x})\varepsilon}) \leq \frac{n}{\log n/\beta}$. Algorithm 2 is $\varepsilon$-differentially private and with probability $1 - \beta$,*

$$|F(v_{\mathsf{out}}) - F(\hat{v})| \leq O\left(\frac{d \log n/\beta}{n\mathsf{GAP}(\boldsymbol{x})\varepsilon} + \frac{1}{n^4}\right).$$

## 4.3 Linear regression

For our final example, we investigate the setting of linear regression where we have data points $(x_i, y_i) \in \mathbb{R}^d \times \mathbb{R}$. Our goal here is to find $\theta \in \Theta$ that minimizes

$$\hat{\theta}_n = \underset{\theta \in \Theta}{\operatorname{argmin}} \; L_n(\theta; \boldsymbol{x}, \boldsymbol{y}) := \frac{1}{2n} \sum_{i=1}^n (\langle \theta, x_i \rangle - y_i)^2.$$

We let $X \in \mathbb{R}^{n \times d}$ has $x_i$ in the $i$'th row, $\Sigma_n = \frac{1}{n} X^T X$, and $\boldsymbol{y} \in \mathbb{R}^d$ denote the vector of $y_i$.

The gradient mechanism (4) have an efficient implementation in this setting. Assuming $\Sigma_n \succ 0$, we get that $\nabla L_n(\theta) = \Sigma_n(\theta - \bar{\theta})$ where $\bar{\theta} = \frac{1}{n}\Sigma_n^{-1}X^T\boldsymbol{y}$. The gradient mechanism has density

$$\pi(\bar{\theta} + \Delta \mid \boldsymbol{x}) \propto e^{-\frac{n\varepsilon}{2L}\|\Sigma_n\Delta\|_2},$$

for $\bar{\theta} + \Delta \in \Theta$ (and Lipschitz constant $L$) which Algorithm 3 samples from (see Appendix F.1). The main difficulty in Algorithm 3 is calculating the non-private estimator $\bar{\theta}$ in step 1 while the remaining steps (for private noise addition) only require sampling from simple distributions and matrix-vector products.

---

**Algorithm 3:** Gradient mechanism for linear regression

---

**Input:** $(x_i, y_i)_{i=1}^n$
1. Calculate $\Sigma_n = \frac{1}{n}X^TX$, $\bar{\theta} = \frac{1}{n}\Sigma_n^{-1}X^T\boldsymbol{y}$;
2. Sample $R \sim \mathsf{Gamma}(d, 1)$, $U \sim \mathsf{Uni}(\mathbb{S}^{d-1})$;
3. Set $\theta_{\mathsf{out}} = \bar{\theta} + \frac{2L}{n\varepsilon}\Sigma_n^{-1} \cdot R \cdot U$ ;
4. **if** $\theta_{\mathsf{out}} \notin \Theta$ **then** go to 3 ;
5. **return** $\theta_{\mathsf{out}}$;

---

The following proposition states the utility and privacy guarantees of Algorithm 3.

**Proposition 4.3.** *For the set of instances $(x_i, y_i)_{i=1}^n$ with $\Sigma_n(\boldsymbol{x}) \succ 0$ and Lipschitz constant $L$, Algorithm 3 is $\varepsilon$-differentially private. Moreover, if $\inf_{\theta \in \mathrm{bd}\,\Theta} \|\theta - \hat{\theta}_n\|_2 \geq \Omega\left(\frac{dL^2}{n^2\varepsilon^2}\,\mathrm{tr}\left(\Sigma_n(\boldsymbol{x})^{-2}\right)\right)$,*

$$\mathbb{E}\left[L_n(\theta_{\mathsf{out}}; \boldsymbol{x}, \boldsymbol{y}) - L_n(\hat{\theta}_n; \boldsymbol{x}, \boldsymbol{y})\right] \leq O\left(\frac{dL^2\,\mathrm{tr}\left(\Sigma_n(\boldsymbol{x})^{-1}\right)}{n^2\varepsilon^2}\right).$$

To appreciate the instance-specific upper bounds of Proposition 4.3, recall that existing private algorithms for empirical risk minimization of $L$-Lipschitz and $\lambda$-strongly convex functions achieve excess loss $\mathbb{E}[L_n(\theta) - L_n(\hat{\theta}_n)] = O(\frac{d^2L^2}{n^2\varepsilon^2\lambda})$ which is minimax optimal in some regimes [5]. In contrast, for natural instances where $\Sigma_n(\boldsymbol{x})^{-1}$ has polynomially decaying eigenvalues $\lambda_j = j^{-\alpha}$ for $\alpha \in (0, 1]$, Proposition 4.3 implies that Algorithm 3 achieves excess loss $\widetilde{O}(\frac{d^{2-\alpha}L^2}{n^2\varepsilon^2})$ which can offer up to $\widetilde{O}(d)$ improvement. Finally, we note that Wang [34]—which focuses on approximate $(\varepsilon, \delta)$-DP—develops private algorithms for linear regression that exhibit good adaptivity to the difficulty of the underlying instance. There exists an extensive prior work on private linear regression and—as this is not the main focus of our work—we refer the reader to [34, 28] for a survey of results.

**Comparison with the smooth sensitivity framework** We conclude the paper with a short comparison of the smooth sensitivity framework and inverse sensitivity mechanisms. While smooth sensitivity mechanisms may not be instance-optimal in many settings, Asi and Duchi [4] show that the inverse sensitivity mechanism is (nearly) instance-optimal for most well-behaved functions and can offer quadratic improvement in sample complexity over smooth sensitivity mechanisms in certain settings. The inverse sensitivity mechanism also outperforms smooth Laplace uniformly for every instance for natural families of sample-monotone functions (see Section 4.3 in [4]). As our development in this paper shows, the approximate versions of the inverse sensitivity mechanism still enjoy similar advantages over smooth mechanisms. Proposition 2.2 shows that—for certain choices of approximations—the approximate inverse sensitivity mechanisms uniformly outperform the smooth Laplace mechanism for every instance. Moreover, the smooth sensitivity framework requires adding noise with heavy-tailed distributions and unbounded moments (such as Cauchy) to preserve $\varepsilon$-differential privacy, in contrast to the approximate inverse sensitivity mechanisms which (depending on the approximation and inverse sensitivity) has noise with exponentially decaying tails, resulting in better high-probability bounds and confidence intervals. The PCA example clearly demonstrates these advantages where the approximate inverse sensitivity mechanism enjoys a factor of $\sqrt{d}$ improvement in sample complexity over smooth sensitivity mechanisms.

## Broader Impact

The substantial growth in data collection and analysis and the increasing awareness for privacy concerns has led to a growing body of work on privacy risks in both academic [16] and industrial settings [17, 3]. Differential privacy [16] has emerged as the standard method for preserving privacy and has enjoyed several applications including in statistical estimation [11], machine learning [5], and game theory [24].

Unfortunately, it is usually challenging to develop private algorithms that achieve satisfactory utility [11]. Therefore, while differential privacy has been successfully deployed in several industrial companies, most applications instantiate a large privacy parameter $\varepsilon$ to achieve acceptable utility, potentially compromising the privacy of users [1].

However, the standard approach in differential privacy to measure the performance of an algorithm is through its (worst case) minimax risk [11]. This—as our theory demonstrates—may be too pessimistic in general and may not capture the correct trade-off between privacy and utility for natural data that arises in real-life. An instance-specific understanding of this trade-off can therefore result in significant improvements in both utility and privacy.

We hope that this work—and instance-optimality in differential privacy in general [4]—can lead to a better understanding of the privacy-utility trade-off of private algorithms for the underlying data at hand. By exploiting the average-case nature of data in real life, we believe that the instance-optimal algorithms we develop can achieve satisfying utility with significantly stronger privacy protections for users.

## Funding Transparency Statement

Funding in direct support of this work: NSF CAREER CCF-1553086, ONR YIP N00014-19-2288, Sloan Foundation, NSF HDR 1934578 (Stanford Data Science Collaboratory), and Stanford DAWN Consortium.

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
