[Supplementary Material]

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

# Appendix

## A   Proofs of Section 2.1

### A.1   Proof of Theorem 1

Using the privacy guarantees of the exponential mechanism [24, Theorem 6], it is enough to prove that $\overline{\mathsf{len}}_f(\boldsymbol{x}; t)$ is 1-Lipschitz. Let $\boldsymbol{x}, \boldsymbol{x}'$ be two neighboring datasets and let $t \in \mathcal{T}$. Assume w.l.o.g. that $\ell = \overline{\mathsf{len}}_f(\boldsymbol{x}; t) \leq \overline{\mathsf{len}}_f(\boldsymbol{x}'; t)$. We need to prove that $\overline{\mathsf{len}}_f(\boldsymbol{x}'; t) \leq \ell + 1$. From the definition of $\overline{\mathsf{len}}$ in Equation (1), we have that $\sum_{i=1}^{\ell} R_i(\boldsymbol{x}) \geq \|t - f(\boldsymbol{x}')\|_p$, so the claim follows from conditions on $R_\ell(\cdot)$ since

$$\|t - f(\boldsymbol{x}')\|_p \leq \|f(\boldsymbol{x}) - f(\boldsymbol{x}')\|_p + \|t - f(\boldsymbol{x})\|_p \leq R_1(\boldsymbol{x}') + \sum_{i=1}^{\ell} R_i(\boldsymbol{x}) \leq \sum_{i=1}^{\ell+1} R_i(\boldsymbol{x}').$$

### A.2   Proof of Proposition 2.1

To prove the claim about privacy, it is enough to show that $R_\ell$ satisfy the conditions of Lemma 1. We clearly have $\mathsf{LS}(\boldsymbol{x}) \leq \overline{\mathsf{LS}}(\boldsymbol{x}) \leq R_1(\boldsymbol{x})$ for all $\boldsymbol{x} \in \mathcal{X}^n$. Moreover, we have that $R_\ell(\boldsymbol{x}) \leq R_{\ell+1}(\boldsymbol{x}')$ for all neighboring datasets $\boldsymbol{x}, \boldsymbol{x}' \in \mathcal{X}^n$ since

$$R_\ell(\boldsymbol{x}) = \sup_{\boldsymbol{x}_1 : d_{\mathrm{ham}}(\boldsymbol{x}, \boldsymbol{x}_1) \leq \ell} \overline{\mathsf{LS}}(\boldsymbol{x}_1) \leq \sup_{\boldsymbol{x}_1 : d_{\mathrm{ham}}(\boldsymbol{x}', \boldsymbol{x}_1) \leq \ell+1} \overline{\mathsf{LS}}(\boldsymbol{x}_1) = R_{\ell+1}(\boldsymbol{x}').$$

### A.3   Proof of Theorem 2

We begin with the exact inverse sensitivity mechanism. Let $C^k = \{t : \mathsf{len}_f^\rho(\boldsymbol{x}; t) = k\}$. The definition of $\mathsf{len}^\rho$ implies that $\mathsf{len}_f^\rho(\boldsymbol{x}; t) = 0$ for $t$ such that $\|t - f(\boldsymbol{x})\|_2 \leq \rho$ and that $\mathsf{len}_f^\rho(\boldsymbol{x}; t) \geq K$ for any $t$ such that $\|t - f(\boldsymbol{x})\|_2 \geq \omega_f(\boldsymbol{x}; K) + \rho$. Thus we have that

$$
\begin{aligned}
P\left(\|M_{\mathrm{inv}}(\boldsymbol{x}) - f(\boldsymbol{x})\|_2 \geq \omega_f(\boldsymbol{x}; K) + \rho\right) &\leq \sum_{k=K}^{n} \mathbb{P}(M_{\mathrm{inv}}(\boldsymbol{x}) \in C^k) \\
&\leq \frac{e^{-K\varepsilon/2} \sum_{k=K}^{n} \int_{s \in C^k} ds}{\int_{s \in \mathcal{T}} e^{-\mathsf{len}_f^\rho(\boldsymbol{x}; t)\varepsilon/2} ds} \\
&\leq e^{-K\varepsilon/2} \frac{\mathrm{Vol}\{t : \|t\|_2 \leq D\}}{\mathrm{Vol}\{t : \|t\|_2 \leq \rho\}} \\
&\leq e^{-K\varepsilon/2} (D/\rho)^d.
\end{aligned}
$$

This gives the first part of the claim.

Now we prove the bounds for the approximate mechanism. First, we notice that the noise added by the approximate mechanism (M.2) satisfies

$$z(\boldsymbol{x}) \coloneqq M(\boldsymbol{x}) - f(\boldsymbol{x}),$$

where $\mathbb{P}(z(\boldsymbol{x}) = z) \propto e^{-k\varepsilon/2}$ for $z \in B^k = \{z \in \mathbb{R}^d : \sum_{i=1}^{k-1} S_i \leq \|z\|_2 < \sum_{i=1}^{k} S_i\}$. Noting that $\|z(\boldsymbol{x})\|_2 \geq \sum_{i=1}^{K} R_i(\boldsymbol{x})$ implies that $z(\boldsymbol{x}) \in B^k$ for $k \geq K$, we get that

$$
\begin{aligned}
\mathbb{P}\left(\|M_{\mathrm{appr}}(\boldsymbol{x}) - f(\boldsymbol{x})\|_2 \geq \sum_{i=1}^{K} R_i(\boldsymbol{x})\right) &= \mathbb{P}\left(\|z(\boldsymbol{x})\|_2 \geq \sum_{i=1}^{K} R_i(\boldsymbol{x})\right) \\
&\leq \sum_{k=K}^{n} \mathbb{P}(z(\boldsymbol{x}) \in B^k) \\
&\leq e^{-K\varepsilon/2} \frac{\mathrm{Vol}\{t : \|t\|_2 \leq nD\}}{e^{-1}\mathrm{Vol}\{t : \|t\|_2 \leq \sum_{i=1}^{1/\varepsilon} R_i(\boldsymbol{x})\}} \\
&\leq e^{-K\varepsilon/2+1}\left(\frac{nD}{\sum_{i=1}^{1/\varepsilon} R_i(\boldsymbol{x})}\right)^d,
\end{aligned}
$$

where the last inequality follows using that the ratio of the volumes of two $\ell_p$-balls with radii $r_1$ and $r_2$ is $(r_1/r_2)^d$.

### A.4  Proof of Proposition 2.2

The claim about privacy follows from identical arguments to the proof of Proposition 2.1. We now prove the claim about utility. We remove $\boldsymbol{x}$ to simplify notation. First, we have that $1 \leq \frac{R_i}{R_1} \leq e^{i\beta}$ from the definition of smooth sensitivity. Thus we have

$$
\begin{aligned}
\mathbb{E}\left[|M(\boldsymbol{x}) - f(\boldsymbol{x})|\right] &= \frac{\sum_{i=1}^{n} e^{-i\varepsilon/2} R_i \sum_{j=1}^{i} R_j}{\sum_{i=1}^{n} e^{-i\varepsilon/2} R_i} \\
&\leq \frac{\sum_{i=1}^{n} e^{-i\varepsilon/2} e^{i\beta} R_1^2 \sum_{j=1}^{i} e^{j\beta}}{\sum_{i=1}^{n} e^{-i\varepsilon/2} R_1} \\
&\leq R_1 \frac{\sum_{i=1}^{n} e^{-i\varepsilon/2} e^{i\beta} \frac{e^{i\beta}}{e^{\beta}-1}}{\sum_{i=1}^{n} e^{-i\varepsilon/2}} \\
&= \frac{R_1}{e^{\varepsilon/8}-1} \frac{\sum_{i=1}^{n} e^{-i\varepsilon/4}}{\sum_{i=1}^{n} e^{-i\varepsilon/2}} \\
&= \frac{R_1}{e^{\varepsilon/8}-1} \frac{e^{\varepsilon/2}-1}{e^{\varepsilon/4}-1} = O\left(\frac{\mathsf{S}^\beta(\boldsymbol{x})}{\varepsilon}\right),
\end{aligned}
$$

where the last equality follows since $\varepsilon = O(1)$.

## B  Proofs of Section 2.2

### B.1  Proof of Lemma 2.1

We begin with some notation. Let $(\boldsymbol{x}, \boldsymbol{y})$ and $(\boldsymbol{x}', \boldsymbol{y}')$ be two neighboring instances, and denote their minimizers by $\hat{\theta}_n$ and $\hat{\theta}'_n$, respectively. We let $D = \mathrm{diam}_2(\Theta)$.

The following lemma bounds the distance between these minimizers.

**Lemma B.1.** *Under the assumptions of Proposition 2.1,*

$$
\left\|\hat{\theta}_n - \hat{\theta}'_n\right\|_2 \leq \frac{2L}{\lambda n}.
$$

To prove Lemma B.1, we first prove the following weaker version.

**Lemma B.2.** *Assume $L_n(\theta; \boldsymbol{x}, \boldsymbol{y})$ is $\lambda$-strongly convex and $L$-Lipschitz. Then*

$$
\left\|\hat{\theta}_n - \hat{\theta}'_n\right\|_2 \leq \frac{L}{\lambda n}.
$$

**Proof** Since $L_n$ is $\lambda$-strongly convex and $L$-Lipschitz, we have

$$\lambda \left\| \hat{\theta}_n - \hat{\theta}'_n \right\|_2^2 \leq \langle \nabla L_n(\hat{\theta}_n; \boldsymbol{x}, \boldsymbol{y}) - \nabla L_n(\hat{\theta}'_n; \boldsymbol{x}, \boldsymbol{y}), \hat{\theta}_n - \hat{\theta}'_n \rangle$$

$$\leq \left\| \nabla L_n(\hat{\theta}_n; \boldsymbol{x}, \boldsymbol{y}) - \nabla L_n(\hat{\theta}_n; \boldsymbol{x}, \boldsymbol{y}) \right\|_2 \left\| \hat{\theta}_n - \hat{\theta}'_n \right\|_2.$$

The claim now follows since $\nabla L_n(\hat{\theta}_n; \boldsymbol{x}, \boldsymbol{y}) = 0$ and $\left\| \nabla L_n(\hat{\theta}'_n; \boldsymbol{x}, \boldsymbol{y}) \right\|_2 \leq \frac{L}{n}$. $\qquad\square$

**Proof** [of Lemma B.1] First, we have that $\nabla L_n(\hat{\theta}_n; \boldsymbol{x}, \boldsymbol{y}) = 0$ and $\left\| \nabla L_n(\hat{\theta}'_n; \boldsymbol{x}, \boldsymbol{y}) \right\|_2 \leq \frac{L}{n}$ since $\ell(\cdot; x_i)$ is $L$-Lipschitz. We split to cases whether $\left\| \hat{\theta}_n - \hat{\theta}'_n \right\|_2 \leq \frac{\lambda}{2H}$. First, if $\left\| \hat{\theta}_n - \hat{\theta}'_n \right\|_2 \leq \frac{\lambda}{2H}$ then we know that the function $L_n(\theta; \boldsymbol{x}, \boldsymbol{y})$ is $\lambda/2$-strongly convex on the set $A = \{\theta : \left\| \hat{\theta}_n - \theta \right\|_2 \leq \frac{\lambda}{2H}\}$. We have $\hat{\theta}'_n \in A$, and therefore Lemma B.2 implies that $\left\| \hat{\theta}_n - \hat{\theta}'_n \right\|_2 \leq \frac{2L}{\lambda n}$.

Now assume that $\left\| \hat{\theta}_n - \hat{\theta}'_n \right\|_2 > \frac{\lambda}{2H}$ and we get a contradiction. Indeed let $\theta_t = (1-t)\hat{\theta}_n + t\hat{\theta}'_n$. For any $0 \leq t \leq 1$ such that $\left\| \hat{\theta}_n - \theta_t \right\|_2 \leq \frac{\lambda}{2H}$, we have that

$$\lambda \left\| \hat{\theta}_n - \theta_t \right\|_2^2 \leq \langle \nabla L_n(\theta_t; \boldsymbol{x}, \boldsymbol{y}), \theta_t - \hat{\theta}_n \rangle$$

$$\overset{(i)}{\leq} \langle \nabla L_n(\theta_1; \boldsymbol{x}, \boldsymbol{y}), \theta_1 - \hat{\theta}_n \rangle$$

$$\leq \frac{L \left\| \hat{\theta}'_n - \hat{\theta}_n \right\|_2}{n},$$

where the third inequality follows from Cauchy-Schwartz inequality since $\theta_1 = \hat{\theta}'_n$ and $(i)$ follows from a monotonicity argument which we explain presently. This implies that $\left\| \hat{\theta}'_n - \hat{\theta}_n \right\|_2 \geq \frac{n\lambda^2}{4LH^2}$ which is a contradiction.

Let us now explain why inequality $(i)$ holds. First we denote $u = \hat{\theta}'_n - \hat{\theta}_n$ and we notice that $\theta_t = \hat{\theta}_n + tu$. Define $g(t) = L_n(\theta_t; x)$ which is convex in $t$. As $g$ is convex with minimizer at $t^\star = 0$, we have $g'(0) = 0$ and $0 \leq g'(t) \leq g'(s)$ for $0 \leq t \leq s$. Therefore we have that $0 \leq \langle \nabla L_n(\theta_t; \boldsymbol{x}, \boldsymbol{y}), u \rangle \leq \langle \nabla L_n(\theta_s; \boldsymbol{x}, \boldsymbol{y}), u \rangle$. Inequality $(i)$ now follows since

$$\langle \nabla L_n(\theta_t; \boldsymbol{x}, \boldsymbol{y}), \theta_t - \hat{\theta}_n \rangle = t \langle \nabla L_n(\theta_t; \boldsymbol{x}, \boldsymbol{y}), u \rangle$$

$$\leq s \langle \nabla L_n(\theta_s; \boldsymbol{x}, \boldsymbol{y}), u \rangle$$

$$= \langle \nabla L_n(\theta_s; \boldsymbol{x}, \boldsymbol{y}), \theta_s - \hat{\theta}_n \rangle.$$

$\qquad\square$

Now we are ready to prove Proposition 2.1. First, Lemma B.1 implies that

$$\left\| \hat{\theta}_n - \hat{\theta}'_n \right\|_2 \leq \frac{2L}{\lambda n}.$$

Therefore as $\nabla^2 L_n(\cdot; \boldsymbol{x}, \boldsymbol{y})$ is $H$-Lipschitz, we get

$$\left\| \nabla^2 L_n(\hat{\theta}_n; \boldsymbol{x}, \boldsymbol{y}) - \nabla^2 L_n(\hat{\theta}'_n; \boldsymbol{x}, \boldsymbol{y}) \right\|_2 \leq \frac{2LH}{\lambda n}.$$

As a consequence we have

$$
\begin{aligned}
&\left| \left\| \nabla^2 L_n(\hat{\theta}_n; \boldsymbol{x}, \boldsymbol{y})(\theta - \hat{\theta}_n) \right\|_2 - \left\| \nabla^2 L_n(\hat{\theta}'_n; \boldsymbol{x}', \boldsymbol{y}')(\theta - \hat{\theta}'_n) \right\|_2 \right| \\
&= \left| \left\| \nabla^2 L_n(\hat{\theta}_n; \boldsymbol{x}, \boldsymbol{y})(\theta - \hat{\theta}_n) \right\|_2 - \left\| \nabla^2 L_n(\hat{\theta}'_n; \boldsymbol{x}', \boldsymbol{y}')(\theta - \hat{\theta}_n) + \nabla^2 L_n(\hat{\theta}'_n; \boldsymbol{x}', \boldsymbol{y}')(\hat{\theta}_n - \hat{\theta}'_n) \right\|_2 \right| \\
&\leq \left\| \nabla^2 L_n(\hat{\theta}_n; \boldsymbol{x}, \boldsymbol{y}) - \nabla^2 L_n(\hat{\theta}'_n; \boldsymbol{x}, \boldsymbol{y}) \right\|_2 \left\| \theta - \hat{\theta}_n \right\|_2 + \left\| \nabla^2 L_n(\hat{\theta}'_n; \boldsymbol{x}', \boldsymbol{y}')(\hat{\theta}_n - \hat{\theta}'_n) \right\|_2 \\
&\leq \frac{2DLH}{\lambda n} + \left\| \nabla L_n(\hat{\theta}_n; \boldsymbol{x}', \boldsymbol{y}') \right\|_2 + O(H \left\| \hat{\theta}_n - \hat{\theta}'_n \right\|_2^2) \\
&\leq \frac{2DLH}{\lambda n} + \frac{L}{n} + O\left( H \left( \frac{2LH}{\lambda n} \right)^2 \right).
\end{aligned}
$$

### B.2 Proof of Proposition 2.3

The claim about privacy is immediate from the exponential mechanism. Let us now argue about the claim for utility. To simplify notation, we let $\Sigma = \nabla^2 L(\hat{\theta}_n; \boldsymbol{x}, \boldsymbol{y})$. We later show (see Section F.1) that to sample from the distribution (5), one can sample $R \sim \mathsf{Gamma}(d, 1)$ and $U \sim \mathsf{Uni}(\mathbb{S}^{d-1})$ and then set $\theta = \hat{\theta}_n + Z$ where $Z = \frac{2\mathsf{GS}_{\mathsf{Hess}}}{\varepsilon} \Sigma^{-1} \cdot R \cdot U$, and finally we accept $\theta$ if $\theta \in \Theta$, otherwise we repeat the process, It is easy to show that for $Z$ we have

$$
\mathbb{E}\left[ \|Z\|_2^2 \right] = \frac{4\mathbb{E}[R^2]\mathsf{GS}_{\mathsf{Hess}}^2}{\varepsilon^2} \mathbb{E}\left[ \|\Sigma^{-1}U\|_2^2 \right] \leq \frac{Cd\mathsf{GS}_{\mathsf{Hess}}^2}{\varepsilon^2} \operatorname{tr}\left( \Sigma^{-2} \right),
$$

for a universal constant $C$. But we need to upper bound $\mathbb{E}\left[ \|Z\|_2^2 \mid \hat{\theta}_n + Z \in \Theta \right]$ as this is the error of the mechanism. To finish the proof, we now prove that for every random variable $W$,

$$
\mathbb{E}\left[ \|W\|_2^2 \mid \hat{\theta}_n + Z \in \Theta \right] \leq 2\mathbb{E}\left[ \|W\|_2^2 \right]. \tag{9}
$$

To this end, we let $\rho^2 = \mathbb{E}\left[ \|Z\|_2^2 \right]$ and define three disjoint sets, $S_1 = \{Z : \|Z\|_2 \leq 2\rho\}$, $S_2 = \{Z : \hat{\theta}_n + Z \in \Theta, Z \notin S_1\}$, and $S_3 = \mathbb{R}^d \setminus (S_1 \cup S_2)$. Clearly these sets are disjoint and the assumptions of the Proposition imply that $S_1 \subseteq \Theta$ and therefore $\Theta = S_1 \cup S_2$. Using conditional expectation and denoting $p_i = \mathbb{P}(Z \in S_i)$, we have that

$$
\mathbb{E}\left[ \|W\|_2^2 \right] = \sum_{i=1}^{3} p_i \mathbb{E}\left[ \|W\|_2^2 \mid Z \in S_i \right].
$$

Noting that $p_1 \geq 1/2$ by Markov inequality, we now get that

$$
\begin{aligned}
\mathbb{E}\left[ \|W\|_2^2 \mid \hat{\theta}_n + Z \in \Theta \right] &= \frac{p_1}{p_1 + p_2} \mathbb{E}\left[ \|W\|_2^2 \mid Z \in S_1 \right] + \frac{p_2}{p_1 + p_2} \mathbb{E}\left[ \|W\|_2^2 \mid Z \in S_2 \right] \\
&\leq 2p_1 \mathbb{E}\left[ \|W\|_2^2 \mid Z \in S_1 \right] + 2p_2 \mathbb{E}\left[ \|W\|_2^2 \mid Z \in S_2 \right] \\
&\leq 2\mathbb{E}\left[ \|W\|_2^2 \right].
\end{aligned}
$$

The claim follows.

## C Proofs of Section 3 (lower bounds)

### C.1 Proofs of Theorem 3

We start with the lower bound for unbiased mechanisms. Fix $\boldsymbol{x} \in \mathcal{X}^n$ and assume towards a contradiction that $\mathbb{E}[\|M(\boldsymbol{x}) - f(\boldsymbol{x})\|_2] \leq \frac{1}{8}\beta_k$ where $\beta_k = c \cdot e^{-2k\varepsilon/d}\omega_f(\boldsymbol{x}; k)$. The definition of $W_f(\boldsymbol{x}; k)$ implies that there exists a $\beta_k/4$ packing, namely $S$, of $W_f(\boldsymbol{x}; k)$ of size at least $m_{\beta_k} \geq (\frac{4c\omega_f(\boldsymbol{x};k)}{\beta_k})^d \geq 4^d e^{2k\varepsilon}$. The definition of $W_f(\boldsymbol{x}; k)$ implies that there is an instance $\boldsymbol{x}'$ such

that $d_{\text{ham}}(\boldsymbol{x}, \boldsymbol{x}') \leq k$ and $f(\boldsymbol{x}') = t$ for every $t \in S$, hence we have a set $A$ of size $m_{\beta_k}$ of datasets $\boldsymbol{x}'$ such that $f(\boldsymbol{x}') \in S$. For every $\boldsymbol{x}' \in A$, we define

$$B_{\boldsymbol{x}'} = \{y : \|y - f(\boldsymbol{x}')\|_2 \leq \beta_k/4\}.$$

We now have that:

$$\mathbb{P}(M(\boldsymbol{x}) \in B_{\boldsymbol{x}'}) \overset{(i)}{\geq} \mathbb{P}(M(\boldsymbol{x}') \in B_{\boldsymbol{x}'})e^{-k\varepsilon} \overset{(ii)}{\geq} \mathbb{P}(M(\boldsymbol{x}') \in B_{\boldsymbol{x}})e^{-k\varepsilon}$$

$$\overset{(iii)}{\geq} \mathbb{P}(M(\boldsymbol{x}) \in B_{\boldsymbol{x}})e^{-2k\varepsilon} \overset{(iv)}{\geq} \frac{e^{-2k\varepsilon}}{2},$$

where $(i)$ and $(iii)$ follow from the definition of differential privacy, $(ii)$ follows since $M$ is $\|\cdot\|_2$-unbiased, and $(iv)$ follows from Markov inequality. As the sets $B_{\boldsymbol{x}'}$ are disjoint for $\boldsymbol{x}' \in A$, we have a contradiction

$$1 \geq \sum_{\boldsymbol{x}' \in A} \mathbb{P}(M(\boldsymbol{x}) \in B_{\boldsymbol{x}'}) \geq \frac{1}{2}m_{\beta_k}e^{-2k\varepsilon} \geq \frac{4^d}{2}.$$

To prove the first part of the claim (i.e., the minimax lower bound), we use similar ideas while starting from the assumption that for every $\boldsymbol{x}$ we have $\mathbb{E}[\|M(\boldsymbol{x}) - f(\boldsymbol{x})\|_2] \leq \frac{1}{8}\beta_k$ where $\beta_k = c \cdot \sup_{\boldsymbol{x}'} e^{-k\varepsilon/d}\omega_f(\boldsymbol{x}'; k)$. We again define a packing $A$ (now with $m_{\beta_k} \geq 4^d e^{k\varepsilon}$) and we get using Markov inequality and the definition of differential privacy that $\mathbb{P}(M(\boldsymbol{x}) \in B_{\boldsymbol{x}'}) \geq \mathbb{P}(M(\boldsymbol{x}') \in B_{\boldsymbol{x}'})e^{-k\varepsilon} \geq \frac{e^{-k\varepsilon}}{2}$. This gives a contradiction similarly to our argument above.

## C.2 Local-minimax lower bound and proof of Theorem 4

First, we start by explaining why $r = \Omega(d/\varepsilon)$ is necessary to exclude trivial mechanisms in the local minimax definition (6). Assume that we choose $r \ll d/\varepsilon$. Then for an instance $\boldsymbol{x} \in \mathcal{X}^n$, consider the trivial constant mechanism that sets $M_{\text{triv}}(\boldsymbol{x}') = f(\boldsymbol{x})$ for every $\boldsymbol{x}' \in \mathcal{X}^n$. Clearly this mechanism is $\varepsilon$-differentially private and its local-minimax risk for $\boldsymbol{x}$ is

$$\sup_{\boldsymbol{x}':d_{\text{ham}}(\boldsymbol{x},\boldsymbol{x}')\leq r} E\left[\|M_{\text{triv}}(\boldsymbol{x}') - f(\boldsymbol{x}')\|_2\right] = \sup_{\boldsymbol{x}':d_{\text{ham}}(\boldsymbol{x},\boldsymbol{x}')\leq r} \|f(\boldsymbol{x}) - f(\boldsymbol{x}')\|_2 = \omega_f(\boldsymbol{x}; r).$$

Our lower bounds of Lemma C.1 on the local minimax risk with radius $r$ show this is the optimal risk—up to constant factors that do not depend on $r$—for $\boldsymbol{x}$ whenever $r \leq d/\varepsilon$. Therefore we need to pick a larger value of $r$ such that the optimal mechanism is not the trivial constant mechanism. Picking $r = C \cdot d/\varepsilon$, our lower bounds are roughly $\sup_{1 \leq i \leq C} e^{-i}\omega_f(\boldsymbol{x}; id/\varepsilon)$ (which is usually maximized at $i = 1$ resulting in $\omega_f(\boldsymbol{x}; d/\varepsilon)$) and so the trivial mechanism does not achieve this for $C$ large enough.

Moreover, when $r \ll d/\varepsilon$, no single mechanism $M$ can be instance-optimal according to this definition: if there exists $M^\star$ such that $\mathbb{E}[\|M^\star(\boldsymbol{x}) - f(\boldsymbol{x})\|_2] \leq O(1)\mathcal{R}(\boldsymbol{x}; r) \leq O(1)\omega_f(\boldsymbol{x}; r)$ for every $\boldsymbol{x} \in \mathcal{X}^n$, then we get that

$$\sup_{\boldsymbol{x} \in \mathcal{X}^n} \mathbb{E}[\|M^\star(\boldsymbol{x}) - f(\boldsymbol{x})\|_2] \leq O(1) \sup_{\boldsymbol{x} \in \mathcal{X}^n} \omega_f(\boldsymbol{x}; r) \ll \sup_{\boldsymbol{x} \in \mathcal{X}^n} \omega_f(\boldsymbol{x}; d/\varepsilon),$$

which contradict the minimax lower bounds of Theorem 3.

Theorem 4 follows from the following lemma by setting $r = d/\varepsilon$.

**Lemma C.1.** *Let the assumptions of Theorem 4 hold. Then for any $r \geq 1$, $\mathcal{R}(\boldsymbol{x}; r) \geq \frac{c}{8} \sup_{k \leq r} e^{-k\varepsilon/d}\omega_f(\boldsymbol{x}; k)$.*

**Proof** The proof follows similar arguments to those we had in the proof of Theorem 3. Assume toward a contradiction that $\mathcal{R}(\boldsymbol{x}; r) \leq \frac{1}{8}\beta_k$ where $\beta_k = c \cdot e^{-k\varepsilon/d}\omega_f(\boldsymbol{x}; k)$ for $k \leq r$. This implies that there exists a mechanism $M$ such that for every $\boldsymbol{x}'$ such that $d_{\text{ham}}(\boldsymbol{x}, \boldsymbol{x}') \leq r$, we have $\mathbb{E}[\|M(\boldsymbol{x}') - f(\boldsymbol{x}')\|_2] \leq \frac{1}{8}\beta_k$. Repeating the arguments of the proof of Theorem 3 with $W_f(\boldsymbol{x}; k)$ for $k \leq r$ proves the claim. $\quad\square$

# D    Proofs and further details of Section 4.1 (mean estimation)

In this section we provide proofs for the claims in Section 4.1 and give our algorithm. We begin by giving the full version of Lemma 4.1, which we prove in Appendix D.2.

**Lemma** (Full version of Lemma 4.1). *Let* $f(\boldsymbol{x}) = [\mathrm{trim}_m(\boldsymbol{x})]_{[a,b]}$. *Then, for any* $t \in [a,b]$, *if* $t \geq \mathrm{trim}_m(\boldsymbol{x})$

$$\mathrm{len}_f(\boldsymbol{x};t) = \min\{k : k \leq m, |t - f(\boldsymbol{x})| \leq \frac{1}{n-2m}\sum_{i=1}^{k}(x_{(n-m+i)} - x_{(m+i)})\} \cup \{m+1\}.$$

*Moreover, if* $t \leq \mathrm{trim}_m(\boldsymbol{x})$

$$\mathrm{len}_f(\boldsymbol{x};t) = \min\{k : k \leq m, |t - f(\boldsymbol{x})| \leq \frac{1}{n-2m}\sum_{i=1}^{k}(x_{(n-m+1-i)} - x_{(m+1-i)})\} \cup \{m+1\}.$$

## D.1    Sampling from the inverse sensitivity mechanism

We describe an algorithm for sampling from the inverse sensitivity mechanism with $\rho > 0$ for the mean estimation problem of Section 4.1. Our goal is to sample from

$$\pi_{M_{\mathrm{inv}}(\boldsymbol{x})}(t) = \frac{e^{-\mathrm{len}^\rho(\boldsymbol{x};t)\varepsilon/2}}{\int_{\mathcal{T}} e^{-\mathrm{len}^\rho(\boldsymbol{x};s)\varepsilon/2}ds}.$$

Algorithm 4 shows how to sample from this distribution using Lemma 4.1. To see this, note that Lemma 4.1 implies that $S_k$ in Algorithm 4 is exactly the set $\{t : \mathrm{len}^\rho(\boldsymbol{x};t) = k\}$. And so all values $t \in S_k$ have the same probability. Moreover, the probability of sampling a value from the set $S_k$ is $\mathrm{Vol}(S_k)e^{-k\varepsilon/2}$ using the definition of the mechanism.

---

**Algorithm 4:** Inverse sensitivity for mean estimation

---

**Input:** $\boldsymbol{x} \in \mathbb{R}^n$, $m$, $\rho$, $a$, $b$

1  Calculate $\hat{x}_t = [\mathrm{trim}_m(\boldsymbol{x})]_{[a,b]}$;
2  Calculate $u_k = \min\left(\rho + \frac{1}{n-2m}\sum_{i=1}^{k}(x_{(n-m+i)} - x_{(m+i)}), b - \hat{x}_t\right)$ for $0 \leq k \leq m$;
3  Calculate $\ell_k = \min\left(\rho + \frac{1}{n-2m}\sum_{i=1}^{k}(x_{(n-m+1-i)} - x_{(m+1-i)}), \hat{x}_t - a\right)$ for $0 \leq k \leq m$;
4  Set $u_{m+1} = b - \hat{x}_t$ and $\ell_{m+1} = \hat{x}_t - a$;
5  Set $S_k = [-\ell_{k+1}, -\ell_k] \cup [u_k, u_{k+1}]$;
6  Sample $k \propto \mathrm{Vol}(S_k)e^{-k\varepsilon/2}$;
7  Sample $z \sim \mathrm{Uni}(S_k)$;
8  **return** $\hat{x} = \hat{x}_t + z$

---

## D.2    Proof of Lemma 4.1

We only prove the case $t \geq \mathrm{trim}_m(\boldsymbol{x})$ as the other one is similar. In this case, to make the value of the trimmed mean $t$ by changing $k$ values, we must change the value of the $k$ smallest samples $x_{(1)}, \cdots, x_{(k)}$ and set their value to $\infty$. Denote the resulting sample by $\boldsymbol{x}'$. The trimmed mean $\boldsymbol{x}'$ is

$$\mathrm{trim}_m(\boldsymbol{x}') = \frac{x'_{(m+1)} + x'_{(m+2)} + \cdots + x'_{(n-m)}}{n - 2m}.$$

We split to two cases. First, if $k \geq m+1$, then we get that $x'_{n-m} = \infty$ and therefore $\mathrm{trim}_m(\boldsymbol{x}') = \infty$. This means that for any $t > \mathrm{trim}_m(\boldsymbol{x})$ we can set suitable new values to $x_{(1)}, \cdots, x_{(k)}$ (instead of $\infty$) such that $\mathrm{trim}_m(\boldsymbol{x}') = t$. Now assume $k \leq m$. In this case, we get that

$$\mathrm{trim}_m(\boldsymbol{x}') = \frac{x_{(m+k+1)} + x_{(m+k+2)} + \cdots + x_{(n+k-m)}}{n - 2m}.$$

The claim follows as we have

$$\mathrm{trim}_m(\boldsymbol{x}') - \mathrm{trim}_m(\boldsymbol{x}) = \frac{\sum_{i=1}^{k}(x_{(n-m+i)} - x_{(m+i)})}{n - 2m}.$$

## D.3 Proof of Proposition 4.1

The privacy guarantees of Algorithm 4 follow from Proposition 3.2 in [4].

To prove the claim about utility, we use the following result from [7] which upper bounds the error of the trimmed mean estimator.

**Lemma D.1** (Bun and Steinke [7], Proposition 10). *Let $x_i \overset{\text{iid}}{\sim} P$ where $P$ has mean $\mu$ and variance $\sigma^2$. Then*

$$\mathbb{E}[(\mathsf{trim}_m(\boldsymbol{x}) - \mu)^2] \leq \frac{n(1 + \sqrt{8m})}{(n - 2m)^2}\sigma^2 = O\left(\frac{m}{n}\right)\sigma^2.$$

*Moreover, if $P$ is symmetric about its mean then*

$$\mathbb{E}[(\mathsf{trim}_m(\boldsymbol{x}) - \mu)^2] \leq \left(1 + O\left(\frac{m}{n}\right)\right)\frac{\sigma^2}{n}.$$

We have that

$$\mathbb{E}[(\hat{x} - \mu)^2] \leq \mathbb{E}[([\mathsf{trim}_m(\boldsymbol{x})]_{[a,b]} - \mu)^2] + \mathbb{E}[z^2]$$
$$\leq \mathbb{E}[(\mathsf{trim}_m(\boldsymbol{x}) - \mu)^2] + \mathbb{E}[z^2],$$

where the second inequality follows since $\mu \in [a, b]$, and so a projection to $[a, b]$ cannot increase error. Thus, given the bound of Lemma D.1, now we only need to upper bound $\mathbb{E}[z^2]$.

We begin with the following lemma for a fixed $\boldsymbol{x}$.

**Lemma D.2.** *Let $\boldsymbol{x} \in \mathbb{R}^n$ and $K \leq m$. Let $L(\boldsymbol{x}) = \max_{1 \leq k \leq K}(u_{k+1} - u_k, \ell_{k+1} - \ell_k)$. Then*

$$\mathbb{E}[z^2] \leq O\left(\frac{L(\boldsymbol{x})^2}{\varepsilon^2}\right) + e^{-K\varepsilon/2}\frac{m(b-a)^2}{\rho}.$$

**Proof**  Let $v_k = u_k + \ell_k$. Then we have that $v_{k+1} - v_k \leq 2L$ for $k \leq K$ and $v_0 \geq \rho$, hence we get

$$\mathbb{E}[z^2] \leq \frac{\sum_{k=0}^{m+1} e^{-k\varepsilon/2}v_k^2(v_k - v_{k-1})}{\sum_{k=0}^{m+1} e^{-k\varepsilon/2}(v_k - v_{k-1})}$$
$$\leq \frac{\sum_{k=0}^{K} e^{-k\varepsilon/2}v_k^2(v_k - v_{k-1})}{\sum_{k=0}^{m+1} e^{-k\varepsilon/2}(v_k - v_{k-1})} + e^{-K\varepsilon/2}\frac{m(b-a)^2}{\rho}$$
$$\overset{(i)}{\leq} O\left(\frac{L^2}{\varepsilon^2}\right) + e^{-K\varepsilon/2}\frac{m(b-a)^2}{\rho}.$$

Inequality $(i)$ follows from similar arguments to the proof of Proposition 4.3 in [4]: let $T \leq K$ be the smallest such that $v_T \geq \frac{L}{\varepsilon}$. If no such $T$ exists, then inequality $(i)$ clearly holds. Note that $v_T \leq \frac{L}{\varepsilon} + L \leq O(\frac{L}{\varepsilon})$. Thus we get

$$\frac{\sum_{k=1}^{K} e^{-k\varepsilon/2}v_k^2(v_k - v_{k-1})}{\sum_{k=1}^{m+1} e^{-k\varepsilon/2}(v_k - v_{k-1})} \leq v_T + \frac{\sum_{k=T}^{K} 2e^{-k\varepsilon/2}k^2L^3}{e^{-T\varepsilon/2}L/\varepsilon} = O\left(\frac{L^2}{\varepsilon^2}\right).$$

$\square$

Now we are ready to finish the proof of Proposition 4.1. We notice that for any $\boldsymbol{x}$

$$L(\boldsymbol{x}) \leq \frac{x_{(n)} - x_{(1)}}{n - m} \leq \frac{2\max_{1 \leq i \leq n}|x_i|}{n - m}.$$

Therefore using standard bound on the expectation of maximum of subgaussian variables (Lemma 45 in full version [7])

$$\mathbb{E}[L(\boldsymbol{x})^2] \leq \frac{4\mathbb{E}[\max_{1 \leq i \leq n} x_i^2]}{(n - m)^2} \leq \frac{8\sigma^2 \log n}{(n - m)^2}.$$

Overall we have that using $K = m$ in Lemma D.2 implies that for a constant $C$,

$$\mathbb{E}[(\hat{x} - \mu)^2] \le \mathbb{E}[(\text{trim}_m(\boldsymbol{x}) - \mu)^2] + \frac{C\sigma^2 \log n}{(n-m)^2 \varepsilon^2} + e^{-m\varepsilon/2} \frac{m(b-a)^2}{\rho}.$$

As $\rho = \frac{\sigma^2}{n^2}$ and $m \le n$, setting $m = \frac{12 \log(n(b-a)/\sigma^2)}{\varepsilon}$ and using Lemma D.1 proves the claim.

## E    Proofs of Section 4.2 (PCA)

Here we prove Proposition 4.2. First, we prove the claim about privacy then we proceed to show the utility analysis.

### E.1    Proof of Proposition 4.2 (privacy)

We only need to prove that $w(\boldsymbol{x}) = \bar{v} + z$ is $\varepsilon$-DP as the claim for Algorithm 2 then follows since post-processing preserves privacy [13, Proposition 2.1]. To this end, first we note that Lemma 4.2 implies that the choice of $R_i(\boldsymbol{x})$ in Proposition 4.2 satisfies the conditions of Theorem 1. Now assume we have two neighboring instances $\boldsymbol{x}, \boldsymbol{x}'$ with leading eigenvectors $u_1, u_2 = -u_1$ and $u_1', u_2' = -u_1'$ respectively and assume without loss of generality $\|u_1 - u_1'\|_2 \le \text{LS}(\boldsymbol{x})$. We get that $w(\boldsymbol{x}) = w_1(\boldsymbol{x}) = u_1 + z$ with probability $1/2$ and $w(\boldsymbol{x}) = w_2(\boldsymbol{x}) = u_2 + z$ otherwise. Similarly we have $w_1(\boldsymbol{x}')$ and $w_2(\boldsymbol{x}')$. Theorem 1 now implies that the densities of $w_1(\boldsymbol{x})$ and $w_1(\boldsymbol{x}')$ are $\varepsilon$-DP (i.e., $\frac{\pi_{w_1(\boldsymbol{x})}(t)}{\pi_{w_1(\boldsymbol{x}')}(t)} \le e^\varepsilon$) and similarly for the densities of $w_2(\boldsymbol{x})$ and $w_2(\boldsymbol{x}')$, therefore by quasi convexity we get that $w(\boldsymbol{x})$ is $\varepsilon$-DP.

### E.2    Proof of Proposition 4.2 (utility)

To facilitate notation, we drop $\boldsymbol{x}$ from our analysis. First, we bound the norm of the noise $z$ that the algorithm adds. We claim that there exists a universal constant $C_1 > 0$ such that the noise $z$ in step 4 of Algorithm 2 has with probability $1 - \beta$

$$\|z\|_2 \le C_1 \left( \frac{d \log n/\beta}{n\text{GAP}(\boldsymbol{x})\varepsilon} + \frac{1}{n^4} \right).$$

Deferring the proof of this, we can now complete the proof of the claim. We assume that $n$ is large enough so that $\|z\|_2 \le 1/2$. Notice that in our setting ($k = 1$) we have that $F(v) = v^T \Sigma v = \|\Sigma^{1/2} v\|_2^2$. Therefore denoting $\lambda = \frac{1}{\|\bar{v} + z\|_2}$ we get that

$$
\begin{aligned}
|F(v_{\text{out}}) - F(\bar{v})| &\le |F(v_{\text{out}}) - F(\bar{v} + z)| + |F(\bar{v} + z) - F(\bar{v})| \\
&= |\lambda^2 - 1| F(\bar{v} + z) + \left| \left\|\Sigma^{1/2}(\bar{v} + z)\right\|_2^2 - \left\|\Sigma^{1/2}\bar{v}\right\|_2^2 \right| \\
&\le |\lambda^2 - 1| \|\bar{v} + z\|_2 + \left\|\Sigma^{1/2} z\right\|_2^2 \left( \left\|\Sigma^{1/2}(\bar{v} + z)\right\|_2^2 + \left\|\Sigma^{1/2}\bar{v}\right\|_2^2 \right) \\
&\le 2|\lambda^2 - 1| + 3 \|z\|_2^2,
\end{aligned}
$$

where we use the fact that $\|z\|_2 \le 1/2$, $\|\bar{v}\|_2 = 1$, and $\|x_i\|_2 \le 1$ so that $\left\|\Sigma^{1/2} u\right\|_2 \le \|u\|_2$ for every $u$. Now we only need to upper bound $|\lambda^2 - 1|$. As $\lambda \le 2$, we have

$$|\lambda^2 - 1| \le 3|\lambda - 1| = 3 \frac{|1 - \|\bar{v} + z\|_2|}{\|\bar{v} + z\|_2} \le 6 \|z\|_2.$$

Therefore overall we have that

$$|F(v_{\text{out}}) - F(\hat{v})| \le 15 \|z\|_2,$$

which proves the claim.

Now we return to prove the claim about the norm of $z$. We use Theorem 2 with $K = \frac{c_1 d \log n/\beta}{\varepsilon}$ to get

$$\mathbb{P}\left( \|z\|_2 \ge \sum_{i=1}^{K} R_i(\boldsymbol{x}) \right) \le \frac{e^{-K\varepsilon/2}}{e} \left( \frac{n\sqrt{2}}{\sum_{i=1}^{1/\varepsilon} R_i(\boldsymbol{x})} \right)^d.$$

Assuming $K \leq n\mathsf{GAP}(\boldsymbol{x})/4$ as we can take $n$ large enough in the assumption of the proposition, we get that $R_i(\boldsymbol{x}) \leq \frac{2C_{\mathsf{pca}}}{n\mathsf{GAP}(\boldsymbol{x})}$ for $i \leq K$ and therefore $\sum_{i=1}^{K} R_i(\boldsymbol{x}) = O(\frac{C_{\mathsf{pca}}d\log n/\beta}{n\mathsf{GAP}(\boldsymbol{x})\varepsilon})$. Since $R_i(\boldsymbol{x}) \geq \frac{C_{\mathsf{pca}}}{n\mathsf{GAP}(\boldsymbol{x})}$, setting $c_1$ large enough we get that

$$\mathbb{P}\left( \|z\|_2 \geq \sum_{i=1}^{K} R_i(\boldsymbol{x}) \right) \leq \frac{e^{-K\varepsilon/2}}{e} \left( \frac{n^2 \mathsf{GAP}(\boldsymbol{x})\sqrt{2}}{C_{\mathsf{pca}}} \right)^d \leq \beta.$$

## F   Proofs of Section 4.3 (linear regression)

### F.1   Sampling from the gradient mechanism

In this section, we show how Algorithm 3 is basically sampling from the distribution of the gradient mechanism. To this end, we show how to sample a vector $t \in \mathbb{R}^d$ with density $\pi(t) = \exp(-\|At\|)$ for a matrix $A \succ 0$. The change of variables $u = At$ and then using rotational symmetry gives that

$$\int \pi(t)dt = \frac{1}{\det(A)} \int \exp(-\|u\|)du = \frac{1}{\det(A)} \int_0^\infty \exp(-r)\mathrm{Vol}_{d-1}(r\mathbb{S}^{d-1})dr$$

$$= \frac{1}{\det(A)}\frac{d\pi^{d/2}}{\Gamma(\frac{d}{2}+1)}\int_0^\infty r^{d-1}e^{-r}dr = \frac{d\pi^{d/2}\Gamma(d)}{\det(A)\Gamma(\frac{d}{2}+1)}.$$

In particular, to sample $T$ with the density $\pi(t) = \exp(-\|At\|)$, we draw $R \sim \mathsf{Gamma}(d,1)$, then $U \sim \mathsf{Uni}(\mathbb{S}^{d-1})$, and set $T = RA^{-1}U$.

Recall that the gradient mechanism has $\pi(t) = \exp(-\|At\|)$ only for $t \in S$ for some set $S \subset \mathbb{R}^d$ and $\pi(t) = 0$ otherwise. Therefore we apply rejection sampling until we get $t \in S$. This shows that Algorithm 3 is sampling from the gradient mechanism.

### F.2   Proof of Proposition 4.3

Similarly to the proof of Proposition 2.3, and letting $Z = \frac{2L}{n\varepsilon}\Sigma_n^{-1} \cdot R \cdot U$ for $R \sim \mathsf{Gamma}(d,1)$, and $U \sim \mathsf{Uni}(\mathbb{S}^{d-1})$, we note that Algorithm 3 sets $\theta_{\mathsf{out}} = \bar{\theta} + Z$ and accepts it if $\theta_{\mathsf{out}} \in \Theta$. We also have

$$L_n(\theta_{\mathsf{out}}; \boldsymbol{x}, \boldsymbol{y}) - L_n(\hat{\theta}_n; \boldsymbol{x}, \boldsymbol{y}) = (\theta_{\mathsf{out}} - \hat{\theta}_n)^T \Sigma_n (\theta_{\mathsf{out}} - \hat{\theta}_n).$$

Since $\hat{\theta}_n \in \mathrm{int}(\Theta)$, we get that $\bar{\theta} = \hat{\theta}_n$ and thus the excess loss of the algorithm is

$$\mathbb{E}\left[ L_n(\theta_{\mathsf{out}}; \boldsymbol{x}, \boldsymbol{y}) - L_n(\hat{\theta}_n; \boldsymbol{x}, \boldsymbol{y}) \right] \leq \mathbb{E}\left[ Z^T\Sigma_n Z \mid \hat{\theta}_n + Z \in \Theta \right].$$

Using inequality (9) in the proof of Proposition 2.3, the claim follows since

$$\mathbb{E}\left[ Z^T\Sigma_n Z \mid \bar{\theta} + Z \in \Theta \right] \leq 2\mathbb{E}\left[ Z^T\Sigma_n Z \right] = \frac{8\mathbb{E}[R^2]L^2}{n^2\varepsilon^2}\mathbb{E}\left[ U\Sigma_n^{-1}U \right] = O\left( \frac{dL^2}{n^2\varepsilon^2}\,\mathrm{tr}\left( \Sigma_n^{-1} \right) \right).$$