[Reviews · NeurIPS 2020]

Review 1

Summary and Contributions: This paper studies data-instance-optimal mechanisms and lower bounds for differential privacy. Specifically, the paper proposed two mechanisms which improve in performance on the inverse sensitivity mechanism, a mechanism proposed recently in [*]. These mechanisms are based on two approximations of the inverse sensitivity mechanism based on local sensitivity, and on the gradient for ERM applications, respectively. The instance-specific lower bound that the paper prove are for vector-valued functions. The paper also show three applications of the inverse sensitivity mechanism, and its approximations. [*] H. Asi and J. Duchi. Near instance-optimality in differential privacy. arXiv:2005.10630 [cs.CR], 2020.

Strengths: Data-instance-optimality is an important direction for differential privacy algorithm design, this paper contributes to it with two interesting algorithms. The paper presents two lower bounds for instance optimal algorithms which improves with respect to previous lower bounds in the fact that they address vector-valued functions.

Weaknesses: Since one of the main motivation of the paper is improving in efficiency, it is unfortunate that in the applications that the paper shows, this is only tangentially discussed. The paper shows only one application each for the two approximated algorithm. A more extensive discussion of other applications and relative bounds would have been more convincing about the usefulness of the algorithms.

Correctness: The different theoretical results look correct.

Clarity: The paper is well structured and easy to follow.

Relation to Prior Work: In general the paper address well the relations to previous works. However, it would have been good to see a discussion about previous work on lower bounds. Moreover, more comparison with the smooth sensitivity framework in the introduction or related works sections would be good - the paper discuss them here and there, while it would be good to have a more in depth discussion all together.

Reproducibility: Yes

Additional Feedback: Can you please clarify in which settings your technique perform better than the smooth sensitivity framework? ------------------------------------------ Comment after author feedback ------------------------------------------ I thank the authors for their answer. My general opinion of the paper is still positive and I think the authors should revise the paper as they discuss in the rebuttal. In particular, concerning my comments I think that they should indeed discuss more the efficiency of their method in the experimental part, and give a clearer comparison with the smooth sensitivity framework.


Review 2

Summary and Contributions: This paper studies instance-optimal DP mechanisms based on the inverse-sensitivity mechanism, building on the recent work of Asi & Duchi (2020). First, due to the intractability of implementing the inverse-sensitivity mechanism in many settings, the authors develop an approximation based on local sensitivity, for which they give a sampling mechanism, and prove utility results for the approximate mechanism. They also develop gradient and hessian-based approximations of the inverse-sensitivity mechanism which are applicable for empirical risk minimization problems. The authors remark that the lower bounds developed in Asi & Duchi (2020) were limited to real-valued functions. The authors extend that work by developing bounds that apply to vector-valued functions. Finally, the paper explores the inverse-sensitivity mechanism and their approximations to the problems of mean estimation, PCA, and linear regression. In each case, they prove upper bounds on the expected error, and compare these bounds to prior results in the literature.

Strengths: This paper continues a very new line of work, developing instance-optimal mechanisms and bounds for differential privacy. This is important both theoretically and practically, as most DP methods used in practice are based on global sensitivity and the Laplace mechanism, which introduce excessive noise to account for worst-case settings. As Asi & Duchi (2020) demonstrate, it is possible for DP mechanisms to perform significantly better when focusing on instance-specific error. Building upon that work, this paper makes instance (near) optimal mechanisms more readily available, and produces new bounds for vector-valued functions. The theoretical results seem to be correct (but were not carefully checked), and provide novel extensions of the work Asi & Duchi (2020) to the setting of vector-valued functions. Furthermore, the approximations to the inverse-sensitivity mechanism allow for practical implementations, and the authors prove results guaranteeing that the approximations are still near-optimal under certain assumptions. The results are interesting and appropriate for NeurIPS

Weaknesses: While the results provide useful approximations and extensions of lower bounds to the vector-valued setting, the paper is limited in its novelty. The work can be seen as an extension of Asi & Duchi (2020), which proposed the novel framework of instance-specific minimax optimality.

Correctness: The results seem to be correct.

Clarity: The paper is a bit difficult to read in general. Section 2.1.1 was the most difficult for me to follow as it includes several complex formulas which are difficult to parse. Additional exposition explaining the significance of various elements in the equations would be helpful. The paper introduces a lot of notation, which can be difficult to keep track of. For instance, when I got to Theorem 2, I could not remember what rho referred to, and it took me a while to find it on line (74). As another example, when reading Section 4.3, I had trouble figuring out what the value L represents, as it is not explained in that section. Eventually, I found that L referred to the Lipschitz coefficient, explained in Section 2.2. There are a few points in the paper where the references to Theorems/Propositions/Lemmas seem incorrect. For instance, on (194) I believe Proposition 2.1 should reference Lemma 2.1, and on (229) Proposition 4 should reference Theorem 4. The read-ability of the paper would be improved if each section were more self-contained. For instance, even if certain notation is defined earlier, a brief sentence reminding the reader what it represents would make the paper easier to follow. Additional exposition explaining the significance of each theoretical result would also be helpful.

Relation to Prior Work: The paper is mostly building off of Asi & Duchi (2020), and does a good job of explaining how their contributions differ from and expand upon that work. The gradient-based mechanism seems very similar to the K-norm gradient mechanism by Reimherr & Awan (2019). The authors should include this citation, and discuss the similarities and differences in section 2.2. Reimherr, Awan "KNG: K-norm gradient mechanism" (2019), NeurIPS.

Reproducibility: Yes

Additional Feedback: I like the approximations to the inverse-sensitivity mechanism, which increase the usability for real-world applications. The utility guarantees for the approximations provide theoretical backing demonstrating that these approximations have similar properties to the instance-optimal mechanism. The lower bounds for vector-valued functions build upon the bounds of Asi & Duchi (2020), and show how the dimension of the statistic affects the rates. Finally, the inclusion of applications to mean-estimations, PCA, and linear regression show that the mechanisms developed in the paper can be applied to problems of interest, with utility guarantees and practical sampling algorithms. I question whether the paper is sufficiently novel. While the results are nice, the paper seems to be, for the most part, building off of Asi & Duchi (2020) and I question whether it provides substantial and novel insights. Additionally, as expressed in the Clarity section, there are some changes the authors can make to improve the readability of the paper. Perhaps, by better explaining the contributions of their paper, they can better persuade that their contributions offer new insights and are more significant than came across in this manuscript. I am currently rating the paper as 6, but look forward to the authors rebuttal. %%%%%%%%%%%%%%%%%%%%%%%%%%%%%%%%%%%%% %%% UPDATE %%%%%%%%%%%%%%%%%%%%%%%%%%%%%%%%%%%%% Thanks to the authors for a detailed rebuttal. The response convinced me of the importance of the approximations developed in the paper, and the extensions to multivariate settings. I also appreciate that the authors intend to make a dedicated section to discuss the connections to smooth sensitivity; I think that this will strengthen the paper. I updated my score to 7.


Review 3

Summary and Contributions: The "inverse sensitivity mechanism" is a generic tool for differentially privately releasing statistics. It approximates the value of a function f on a sensitive dataset x as follows. For each item t in the codomain of f, it assigns t a score which is the minimum number of items in x which need to change to produce an input x' such that f(x') = t. Each t is then sampled with probability inverse exponential in its score (i.e., using the "exponential mechanism"). Asi and Duchi recently conducted a general study of variants of the inverse sensitivity mechanism, showing that for one-dimensional statistics, they are instance-optimal relative to the class of "unbiased" private mechanisms. However, they left the questions of efficient implementation and instance-optimality for higher-dimensional statistics largely unexplored. This work makes three classes of contributions. First, it describes two "approximate" variants of the inverse sensitivity mechanism that can be computed efficiently using local sensitivities and gradient descent. Second, it proves instance-optimality for vector-valued functions relative to a generalization of unbiased mechanisms. And finally, it considers several concrete applications to mean estimation, PCA, and linear regression.

Strengths: While inverse sensitivity mechanisms have been studied in a number of works, there aren't too many settings in which it's actually efficiently computable. This paper makes nice progress toward showing how it can be used in general settings to get state-of-the-art accuracy bounds. While I am not certain how interesting the specific notions of instance-optimality studied are (see below), I think it's an interesting objective to shoot for and this work provides guidance for how to define instance-optimality and calibrate expectations for what's achievable.

Weaknesses: It's not clear to me that the notion of unbiasedness of mechanisms is such a natural or desirable constraint. For example, suppose f is the one-dimensional median function and consider the dataset consisting of: k copies of 0, 1 copy of 3, and k copies of 4. Then the ball B_2 of radius 2 around 2 contains the entire dataset, so an approximate median should land in here with very high probability. Meanwhile, the ball B_3 of radius 2 around 3 excludes nearly half of the dataset but includes a lot of points outside the range of the data. Unbiasedness requires that an an approximate median be more likely to land in B_3 than in B_2. Similarly, the definition of local minimax optimality seems to exclude some useful mechanisms from the benchmark. For example, fix a packing P of datasets in X^n that are all at distances t from each other. Let f be the function that, on input a dataset x, outputs the point in the packing that is closest to x. If t is at least ~log(1/delta)/eps and x is a point in the packing, then propose-test-release can be used to exactly output f(y) for every y in a neighborhood of size, say, t/2 of x. But such a mechanism would be ruled out in any dimension d larger than log(1/delta). (I realize the focus of the paper is on pure DP, so maybe the bound of d/eps is exactly to make the analog of this mechanism for discrete d-dimensional output spaces permissible.)

Correctness: I did not check the proofs in detail, but the techniques and proof ideas look correct.

Clarity: The paper is clearly written.

Relation to Prior Work: Prior work is mostly well-discussed. Some additional comparison could be done to the extensive prior work on linear regression; see, e.g., http://auai.org/uai2018/proceedings/papers/40.pdf for a retrospective and the references therein.

Reproducibility: Yes

Additional Feedback: EDIT: I thank the authors for their feedback and am satisfied with how they addressed my comments. As a technical comment, lower bounds via fingerprinting codes seem unlikely to achieve a log(1/delta)/eps kind of the dependence. The reason is that they don't distinguish between approximate DP and concentrated DP, whereas getting a log(1/delta)/eps upper bound inherently makes use of a delta probability of privacy failure.


Review 4

Summary and Contributions: Update after rebuttal: I'm satisfied with the authors response to the reviews and leave my score at 7. ================= The paper presents approximations to inverse sensitivity, either based on local sensitivity or on loss function gradients, along with instance-specific lower bounds and some application examples.

Strengths: - topic highly relevant to the NeurIPS community - clearly written and explained - to the extent of my knowledge, good grounding in recent work

Weaknesses: I didn’t find any strong weaknesses. One general criticism that applies to many DP-theory papers, and that could be applied here, is lack of experimental validation/proof of concept. Having an algorithm with attractive bounds is great, but not that relevant if people don’t actually use DP (in other areas, I think there is a stronger argument for theory for the sake of understanding, but I haven’t seen it made convincingly for DP). It would be helpful in this regard if papers like this one that propose an algorithm would also provide at least a basic implementation and test on the use cases they consider. However, I don’t think that this is core to the message of the paper, and acknowledge that it would put a strain on the page limit.

Correctness: Yes, as far as I can tell.

Clarity: I found the paper pleasantly clear and easy to read. I will make some small comments in the “additional feedback” box.

Relation to Prior Work: Yes.

Reproducibility: Yes

Additional Feedback: Clarity comments: - In Theorem 1, $x$ and $l$ are a bit overloaded ($l$ having been used frequently in the previous paragraph). It would be easier to parse if the naming was a bit more diverse. - Maybe differentiate between basic inverse sensitivity mechanism and smoothed inverse sensitivity mechanism. It was a hiccup for me to see Theorem 1 referring to “the inverse sensitivity mechanism”, but requiring smoothing. - Proposition 2.2: What does $\varepsilon = O(1)$ mean? Where can $\varepsilon$ vary? - Section 2.2 was the least clear part of the paper to me. It felt a bit rushed, after Sec 2.1 which was very nicely explained. - Proposition 2.3: “Let [...] satisfy the assumptions of Proposition 2.1.” It’s not immediately clear how those assumptions transfer. Maybe spell them out again? - Theorem 3: I would add the 2-superscript in the modulus of continuity. One more general comment: It doesn’t seem surprising that minimax bounds are not representative of the hardness of well-behaved problems, so I don’t think it needs to be mentioned as often as it is.

[Author Response · NeurIPS 2020]

We thank the reviewers for their valuable feedback, which will improve the paper. We will address all typos, errors, and clarity recommendations the reviewers suggest, turning now to the main concerns of each reviewer.

**Response to reviewer 1:**

Regarding the reviewer's comments about applications, we chose to limit the number of applications to three because of space limitations. However, we note that more applications are possible; for example we may consider any setting where a smooth sensitivity algorithm has been applied where our algorithms will offer similar advantages (e.g. statistics on graphs, Ullman and Sealfon 2019). We also note that all of our algorithms have relatively simple noise distributions with efficient implementation and so, as requested by the reviewer, we will add a more detailed explanation that explains the efficiency of our algorithms for each application we consider.

The reviewer recommends clarifying the comparison of our algorithms to the smooth sensitivity framework, which we sprinkled in different sections throughout the paper (probably haphazardly). We will add a unified discussion that explains the differences between the two frameworks and the advantages of the inverse sensitivity approach. Briefly, the smooth sensitivity algorithms may not be instance-optimal (as Asi & Duchi 2020 show) and require noise distributions with heavy tails (e.g. Cauchy, which has unbounded variance), in contrast to our mechanisms. This yields worse concentration—and hence performance—as, for example, our results for PCA demonstrate. Smooth sensitivity algorithms also require computing the smooth sensitivities, which may be challenging (for example, in linear regression).

As requested, we will add a discussion about related work on lower bounds for private mechanisms.

**Response to reviewer 2:**

For the reviewer's main comment on the contributions of this paper with regard to Asi & Duchi 2020, we believe that the approximation frameworks we develop in this paper significantly improve the applicability and scope of instance-optimal mechanisms. Indeed, the mechanisms of Asi & Duchi do not have efficient implementations for general functions and so currently apply only to limited settings. Our approximations, which rely only on the local sensitivities that are usually used in standard private mechanisms, open the door to several new applications of inverse sensitivity algorithms that enjoy instance-optimal behavior. Our applications provide a great example of this: the algorithms of Asi & Duchi 2020 do not have efficient implementation for these, in contrast to our algorithms, which have simple and efficient implementations. Moreover, the work of Asi & Duchi, while an important and interesting step to measure (instance) optimality in private learning and estimation, is essentially limited to 1-dimensional functionals, as it does not establish instance-optimal bounds for vector-valued functions or provide a family of mechanisms for such functions. Such general (vector-valued) functions are the main focus of this submission.

We thank the reviewer for bringing our attention to the Reimherr & Awan's K-norm mechanism (2019), which certainly is relevant, and we will add it to related work, as we were unaware of it. We briefly mention a few differences: their work provides asymptotic utility analyses, without finite sample guarantees on the performance of their algorithms, and they propose an approximate implementation of their mechanisms using an MCMC procedure without providing privacy guarantees for the implementation, which (in our view) is an important component of any putative privacy-preserving algorithm. We will discuss this work more carefully in the final version.

We will adopt all of the the reviewer's suggestions to improve the clarity of the paper in the final version.

**Response to reviewer 3:**

The reviewer discusses limitations of our first notion of instance-optimality against unbiased mechanisms by presenting an example where unbiasedness may not be desirable. Such weaknesses motivate the second notion of instance-optimality we consider, based on local minimax complexity. This is similar to classical statistics (e.g. Le Cam's local asymptotic normality), where local-minimax lower bounds alleviate the weaknesses of lower bounds for unbiased estimators. Many standard mechanisms (including smooth sensitivity algorithms and the mechanisms in this paper), satisfy our definition of unbiasedness, motivating a desire for instance-lower bounds for this large (and important) family of mechanisms.

We also agree with the reviewer's comment on the local-minimax definition as our main focus in this paper is primarily on $\varepsilon$-differential privacy. Extending this appropriately to $(\varepsilon, \delta)$-DP is an interesting future question, which will likely require modifying the definition (especially the radius) and relying on lower bounds via fingerprinting codes (e.g. the work of Steinke and Ullman).

We will also add a discussion about related work for linear regression as the reviewer requests.

**Response to reviewer 4:**

We thank the reviewer for the comments on clarity and will address these in the final version of the paper.

[Meta-Review · NeurIPS 2020]

A very solid contribution to a very new line of work initiated by Asi & Duchi (2020) on instance-optimal mechanisms in DP. Where many popular mechanisms in DP such as Laplace, Gaussian leverage global sensitivity as a worst-case calibration of privacy-producing randomisation, Asi & Duchi developed the inverse-sensitivity mechanism as an approach to reducing noise from worst case to instance specific. The reviewers and this AC appreciated the paper's contributions to this thread. A local sensitivity approximation to the (sometimes intractable) inverse-sensitivity mechanism potentially broaden the practicality of these ideas, with utility analysis and lower bounds rounding out the theoretical treatment nicely. Focus on vector-valued functions continues in the vein of making the general approach more versatile. For these reasons of potential practicality, I felt like any limited novelty in proof techniques as highlighted by reviews are not of significant concern. I encourage the authors to adopt the key reviewer suggestions for improvement in the camera ready: discussing efficiency in the experiments to lend credence to the prime motivation of the paper; improvements to accessible exposition; a new section expounding connections to smooth sensitivity; and further discussion on the motivation of unbiasedness as suggested by R3.